# Common volume satellite studies of polar mesospheric clouds with Odin/OSIRIS tomography and AIM/CIPS nadir imaging

Lina Broman[1], Susanne Benze[1], Jörg Gumbel[1], Ole Martin Christensen[1], Cora E. Randall[2]

[1]Department of Meteorology, Stockholm University, Stockholm, Sweden

5    [2]Laboratory for Atmospheric and Space Physics and Department of Atmospheric and Oceanic Sciences, University of Colorado Boulder, Boulder, CO, USA

*Correspondence to*: Lina Broman (lina.broman@misu.su.se)

**Abstract.** Two important approaches for satellite studies of Polar Mesospheric Clouds (PMC) are nadir measurements adapting phase function analysis and limb measurements adapting spectroscopic analysis. Combining both approaches enables 10    new studies of cloud structures and microphysical processes but is complicated by differences in scattering conditions, observation geometry, and sensitivity. In this study, we compare common volume PMC observations from the nadir viewing Cloud Imaging and Particle Size instrument (CIPS) on the AIM satellite and a special set of tomographic limb observations from the Optical Spectrograph and InfraRed Imager System (OSIRIS) on the Odin satellite performed during 18 days for the years 2010 and 2011 and the latitude range 78°N to 80°N. While CIPS provides preeminent horizontal resolution, the OSIRIS 15    tomographic analysis provides combined horizontal and vertical PMC information. This first direct comparison is an important step towards co-analyzing CIPS and OSIRIS data, aiming at unprecedented insights into horizontal and vertical cloud processes. Important scientific questions on how PMC lifecycle is affected by changes in humidity and temperature due to atmospheric gravity waves, planetary waves and tides can be addressed by combining PMC observations in multiple dimensions. 2- and 3-dimensional cloud structures simultaneously observed by CIPS and tomographic OSIRIS provide a useful 20    tool for studies of cloud growth and sublimation. Moreover, the combined CIPS/tomographic OSIRIS dataset can be used for studies of even more fundamental character, such as the question of the assumption of the PMC particle size distribution.

We perform the first thorough error characterization of OSIRIS tomographic cloud brightness and cloud ice water content (IWC). We establish a consistent method for comparing cloud properties from limb tomography and nadir observations, 25    accounting for differences in scattering conditions, resolution and sensitivity. Based on an extensive common volume, and a temporal coincidence criterion of only 5 minutes, our method enables a detailed comparison of PMC regions of varying brightness and IWC. However since the dataset is limited to 18 days of observations this study do not a comparison of cloud frequency. The cloud properties of OSIRIS tomographic dataset are vertically resolved, while the cloud properties of the CIPS dataset is vertically integrated. To make these different quantities comparable, the OSIRIS tomographic cloud properties cloud 30    scattering coefficient and Ice mass density (IMD) have been integrated over the vertical extent of the cloud to form cloud albedo and IWC that is the same quantity as CIPS cloud products. We find that the OSIRIS albedo (obtained from the vertical integration of the primary OSIRIS tomography product, cloud scattering coefficient) shows very good agreement with the primary CIPS product, cloud albedo with a correlation coefficient of 0.96. However, OSIRIS systematically reports brighter clouds than CIPS and the bias between the instruments (OSIRIS - CIPS) is 3.4e-6 sr$^{-1}$ (±2.9e-6 sr$^{-1}$) on average. The OSIRIS 35    tomography IWC (obtained from the vertical integration of IMD) agrees well with the CIPS IWC, with a correlation coefficient of 0.91. However, the IWC reported by OSIRIS is lower than CIPS, and we quantify the bias to -22 g km$^{-2}$ (±14 g km$^{-2}$) on average.

# 1 Introduction

Polar mesospheric clouds (PMCs) are the highest and coldest clouds of the Earth's atmosphere. During daylight conditions, PMCs are too faint to be observed by the naked eye, but during twilight, when the lower atmosphere lies in earth's shadow, PMCs become visible against the dark sky when they efficiently scatter light from the sun under the horizon. The striking appearance of a silvery shining veil against the darker summer night sky has also given them the name Noctilucent Clouds (NLCs). Despite the remarkably dry environment of the summer polar mesosphere of only a few ppm of water, PMCs can form because the wave-driven residual meridional inter-hemispheric circulation (Lindzen, 1973; Garcia et al., 1985) causes upwelling of air from the stratosphere, and the rising motion over the summer pole leads to strong adiabatic cooling of the summer polar mesospheric region (Fritts and Alexander, 2003). Resulting mesopause temperatures of typically 130-150 K (Lübken, 1999) make it possible for water vapor to condense on cloud nuclei, forming visible PMCs. During the last two decades, extensive research has contributed to knowledge about PMC formation and composition. It is now well established that the clouds consist of water ice that has nucleated onto meteoric smoke material (Hervig et al., 2001, 2009), and that nucleation occurs in bursts at the mesopause (Megner, 2011; Kiliani et al., 2013). The ice particles grow in size by condensation of the surrounding water vapor and sediment when they grow large enough and become visible when they reach sizes above typically 20-30 nm (Rapp et al., 2002; Rapp and Thomas, 2006). Eventually, they descend into a sub-saturated air-mass and sublimate.

Ever since the first observations of these night-shining clouds over 130 years ago (Backhouse, 1985; Jesse, 1885; Leslie, 1985), the clouds have been studied extensively and continue to be a major subject of middle atmospheric research. This is related to the fact that PMCs are very sensitive to changes in temperature and water vapor, and therefore serve both as tracers for long-term changes of the middle atmosphere where direct observations are limited (DeLand et al., 2006; DeLand and Thomas, 2015) and tracers for the dynamics of the polar summer mesosphere (e.g., Fritts et al., 1993). PMCs are frequently used as a tool to study wave activity in the mesosphere (Witt, 1962). Gravity waves excited at lower altitudes that propagate through the PMC layer alter the cloud microphysics and leave behind a detectable footprint, which manifests as a variation in cloud brightness or cloud height that can be observed both using ground-based instruments and satellites. Propagating gravity waves create a downward/upward motion of the PMC layer, and the adiabatic heating and cooling associated with regions of downward and upward motion will cause sublimation and growth of the cloud ice. The resulting structures in the cloud layer can be used to detect and quantify gravity waves that are present in the mesopause region (Thurairajah et al., 2017; Hart et al., 2018). Gravity wave signatures from PMC albedo variations observed by the nadir Cloud Imaging and Particle Size (CIPS) experiment on the Aeronomy of Ice in the Mesosphere (AIM) satellite (Russel et al., 2009) have for example been used to investigate both small scale-gravity waves (Chandran et al., 2009; Taylor et al., 2011) and large-scale gravity waves (Chandran et al., 2010; Zhao et al., 2015). Additionally, CIPS has proven an excellent tool to study various types of cloud structures, such as voids, bands, ice rings, and fronts, that are connected to the complex small-scale dynamics present in the summer mesopause region (Thurairajah et al., 2013).

Extensive research has led to an increased knowledge on the climatology of PMCs. A number of recent studies suggest that PMC occurence frequency has increased in recent decades (e.g. Hervig et al, 2016; Lübken et al., 2018; and references therein). DeLand and Thomas (2015) show an increase of PMC Ice water content (IWC) at latitudes between 50° and 82° both in the northern hemisphere and the southern hemisphere. Fiedler et al. (2016) report on similar PMC trends from lidar observations at 69°N. In their study, they analyse a 22-year PMC dataset and confirm that PMCs are becoming brighter and occur more often. Suggestions that observed changes in PMC properties over time are connected to the increase of methane and carbon dioxide in the troposphere (Thomas et al., 1989; Thomas and Oliviero, 2001) are still under debate. Recently, Lübken et al. (2018) presented model studies of the antropogenic effect on PMCs at mid-latitudes by simulating the increased amounts of

carbon dioxide and water vapor in the mesosphere since the beginning of the industrialization. Their study suggests that increased amounts of water vapor (due to increased methane) is the main reason for the increasing visibility of PMCs at mid-latitudes in the modern era, and that a cooling of the mesosphere is only of secondary importance.

The geographical extent and variation of the PMC layer are monitored continually by both ground-based instruments and satellites that each possess their advantages and disadvantages. For example, NLC camera networks and ground-based lidars possess the advantage of being able to observe the clouds with very high temporal resolution. NLC camera networks can capture the horizontal evolution of a geographically limited region of the PMC layer. Numerous ground-based studies from NLC camera networks have led to remarkable insight into the complexity of spatial structures of the clouds, planetary wave

influence on NLC occurrence frequency, gravity wave effects on the NLC layer and cloud height measurement (e.g., Dalin et al., 2016, 2015; Kirkwood and Stebel, 2003; Witt, 1962). Ground-based lidar systems on the other hand possess the unique capacity to monitor the vertical structure of the cloud with resolution as fine as 30 sec and 50 m (Baumgarten et al., 2012) and give detailed insight into the vertical structure and microphysical processes of a cloud as well as climatology and long-term changes of PMCs (Baumgarten et al., 2012; Fiedler et al., 2016; Kaifler et al., 2013; Ridder et al., 2017; Shibuya et al., 2017;

Suzuki et al., 2013). Ground-based instruments alone, however, lack the ability to monitor the longitudinal and latitudinal variations. Polar-orbiting satellites, on the other hand, are able to provide information about the climatology and global distribution of PMCs (e.g., Bailey et al., 2015; Hervig et al., 2016, Xiao et al., 2016) as well as detailed information about horizontal variations of cloud parameters and wave effects on cloud microphysics (e.g., Rusch et al., 2016; Hart et al., 2018). Each satellite geometry has its limitations: nadir pointing instruments observe horizontal cloud structures but can provide no

or limited information about vertical structures (Hart et al., 2018). Limb scanning instruments provide information on vertical cloud structures but have only coarse horizontal resolution with gaps between observations. Due to instrument design, limb viewing instruments observe PMC as a function of *tangent* altitude, not the real altitude. As a consequence, the PMC retrieval cannot separate low-lying clouds from clouds in the foreground or in the background. The PMC retrieval for the limbviewing Optical Spectrograph and Infrared Imager System (OSIRIS) (Llewellyn et al., 2004) on the Odin satellite (Murtagh et al.,

2002) assumes that the PMC layer is spatially homogeneous along the instrument line of sight (LOS) for the normal limb scans. This assumption is clearly a simplification, and numerous observations from nadir viewing satellites, lidar and ground-based cameras (e.g., Baumgarten et al., 2012; Dalin et al., 2016; Thurairajah et al., 2013; Witt, 1962) have shown that the PMC cloud layer is highly structured both in horizontal and vertical extent.

To overcome the limitation from the assumption of a spatially homogeneous cloud layer and to enable the retrieval of clouds as a function of actual height instead of tangent height, the application of a two-dimensional tomographic retrieval for OSIRIS has been presented by Hultgren et al. (2013). The tomographic approach for OSIRIS retrievals (described in Section 2) provides a tool to study both vertical and horizontal variations of cloud microphysical properties on a local scale, which is useful for detailed studies of cloud growth and destruction (Christensen et al., 2016; Megner et al., 2016). An early attempt to apply a

direct (tomographic) retrieval for a limb viewing instrument was investigated almost two decades ago. The first approach was carried out by Livesey and Read (2000), who developed a tomographic 2D retrieval for the Microwave Limb Sounder (MLS) instrument on the Aura satellite. A tomographic reconstruction of atmospheric constituents and gases can be performed using multiple exposures from multiple viewing angles. The tomographic technique has previously been used to study middle atmospheric phenomena other than PMCs, for example gravity wave activity in airglow in the Mesopause region (e.g., Nygren

et al., 2000; Song, 2017), and stratospheric mesoscale gravity waves (Krisch et al., 2017; 2018). More recently, Hart et al. (2018) successfully demonstrated the first application of using a tomographic technique to CIPS on AIM. They were able to project a novel 2D PMC surface map with gravity wave signatures, providing valuable information on wave characteristics such as wave amplitude and dominant horizontal wavelengths. For the OSIRIS instrument, Degenstein et al (2003, 2004) first

showed the possibility to retrieve both vertical and horizontal structures from a series of limb images taken from the OSIRIS infrared imager and were able to map volume emission rates of the oxygen infrared atmospheric band.

In this paper, we perform a detailed common volume study of PMC cloud brightness and Ice mass density (IMD) from the Odin OSIRIS tomographic retrievals and coincident PMC observations from CIPS on AIM. The occasions for Odin's special tomographic scans were chosen to coincide both in time and space with the CIPS instrument. A comparison of the two instruments is therefore ideally suited for instrument validation and the combination of the two datasets will be valuable in future studies of cloud-wave interaction, studies on particle sizes as well as studies on how the retrieved clouds properties are affected by cloud inhomogeneities. Many scientific questions about the PMC lifecycle are connected to the 2- or 3-dimensional structure of the clouds. Important such questions concern e.g. the effect of gravity waves or dynamical instabilities on the growth, sublimation or appearance of the clouds. Combined observations by (horizontally resolved) nadir instruments and (vertically resolved) limb instruments have a large potential of addressing such multi-dimensional questions. This is true in particular if the datasets involve tomographic analysis, as in the case of the OSIRIS data utilized here.

Taking into account that the satellites have different viewing geometry, resolution and sensitivity, we analyze cloud brightness and the cloud ice in the CV and perform a detailed error analysis. One advantage of comparing tomographic OSIRIS observations to CIPS observations is that both instruments measure scattered radiance, although OSIRIS measures with limb-viewing geometry and CIPS uses nadir-viewing geometry. Another advantage is that the same assumption regarding the mathematical shape of the particle size distribution, namely a Gaussian distribution, is used in both the OSIRIS and the operational CIPS v4.2 retrieval.

The specific aims of this satellite comparison study are:
1. Perform the first thorough error characterization of the Odin OSIRIS tomographic dataset.
2. Validate the tomographic retrieval and error characterization by comparing PMC albedo and IWC from the Odin/OSIRIS retrievals and AIM/CIPS PMC retrievals.
3. Establish a consistent method for comparing cloud properties from a limb sounding tomographic data set to a nadir viewing instrument.
4. Produce a combined dataset of Albedo and IWC that will facilitate future studies of the PMC lifecycle and PMC particle sizes.

This study focuses on comparing the albedo and IWC between the instruments. A future goal is to produce a combined dataset that can be used to study for example more fundamental issues such as the assumptions of PMC the size distribution of PMCs, an assumption that has been questioned in the past. Each instrument used alone can only provide either very fine horizontal resolution (CIPS) or vertical/coarse horizontal resolution (tomographic OSIRIS). However, when combined in an efficient way, OSIRIS can provide vertical information on cloud structures such as double cloud layers or voids, ice distribution at different altitude levels, and information about the existence of particles of different sizes on different altitude levels that can complement the high horizontal resolution of the clouds from CIPS. Additionally, the combined dataset can be used to investigte how waves (inferred from albedo variations in CIPS) affect the cloud lifetime and how nucleation/sublimation processes that affect the vertical distribution of cloud properties (inferred from a vertical cross section from OSIRIS)

The paper is structured as follows: First, in Section 2, the OSIRIS tomographic technique is introduced together with a thorough error characterization of the OSIRIS tomography PMC scatter coefficient and IMD.  Additionally, this section describes the

CIPS PMC dataset and known uncertainties of cloud albedo and IWC. In Section 3, the method used for the instrument comparison is described, including a discussion of the challenges in making tomographic limb and nadir observations consistent. In Section 4, the results of the comparison are presented and discussed. Section 5 provides the conclusions.

## 2 Datasets

### 2.1 Odin OSIRIS

The Swedish-led Odin satellite (Murtagh et al., 2002) was launched on February 20, 2001, into an almost Sun-synchronous polar orbit at 600km with ascending node near 18:00 local solar time. The Odin mission began as a joint project between aeronomy and astronomy with the primary focus of the aeronomic part of the mission on coupling processes in the atmosphere, better understanding of ozone variation and processes in the middle atmosphere, and processes that govern PMC formation and mesospheric variability. Odin carries two instruments, the Sub-Millimeter Radiometer (SMR) (Urban et al., 2007) and OSIRIS (Llewellyn et al., 2004). OSIRIS consists of an atmospheric limb-scanning spectrometer and an infrared imager with the ability to measure vertical profiles of atmospheric trace gases and ice layers in the middle atmosphere (Karlsson and Gumbel, 2005). The Odin instruments scan the limb of the atmosphere in the forward direction as the satellite nods up and down while moving in its polar orbit. The OSIRIS spectrometer observes scattered sunlight as limb radiance in the wavelength range from 275-810 nm with a spectral resolution of about 1 nm. Odin can be operated in different modes that regulate the vertical resolution and altitude region depending on the species of interest. In a standard stratospheric/mesospheric mode, the satellite scans the atmosphere typically from 7 to 107 km. In 2010, a so-called tomographic mode was developed, scanning only PMC altitudes between typically 70 and 90 km. Since 2010, Odin has continued to perform regular observations in this tomographic mode for selected orbits both in the northern hemisphere and in the southern hemisphere up until the present. In the tomographic mode, the distance between subsequent scans is shorter, which increases the horizontal sampling compared to the normal mode. The extended number of lines of sight through a cloud volume produces sufficient information to tomographically retrieve two-dimensional distributions of cloud scattering coefficient as a function of height and horizontal distance. The tomographic algorithm inverts the observed limb radiance into an estimate of the retrieved local scattering coefficient, a measure of the cloud brightness. The tomographic retrieval is based on the Multiplicative Algebraic Reconstruction Technique (MART) developed by Lloyd and Llewellyn (1989) and further developed and adapted to OSIRIS data by Degenstein et al. (1999; 2003; 2004). It was described in detail by Hultgren et al. (2013) and Hultgren and Gumbel (2014). MART is based on maximum probability theory that solves the problem on a ray by ray basis until the retrieval converges.

The limb radiance is measured by Odin as a function of tangent altitude, while the tomography retrieves the local scattering coefficient as a function of actual altitude and angle along orbit (AAO). AAO runs from 0 to 360° and simply denotes the position of the tangent point along the satellite orbit, starting from 0° when the satellite is crossing the equator, increasing to 90° at the northernmost position, etc. One important advantage of the tomographic retrieval is that cloud brightness can be expressed as a local feature. This solves the problem of ordinary limb retrievals that clouds in the fore- and background cannot be distinguished from low-altitude clouds (Karlsson and Gumbel, 2005). The OSIRIS across-track field-of-view (FOV) through the PMC layer is not constant since it depends on the satellite altitude and the distance from the satellite to the limb, which varies with tangent altitude. The viewing angle of OSIRIS is 35.8 arc minutes across track, which yields a value close to 30 km for the FOV width at the tangent altitude in the PMC region for 2010 and 2011, and this is the value that will be used in this study. Over the length of one Common Volume (CV) element (the reader is referred to Fig. 3 for a schematic of the common volume element), the across-track width of the FOV changes by 2%, or 600 m. This is largely negligible in comparison with the spatial extent of one CIPS pixel and therefore not taken into account in this study.

The tomographic retrieval uses a grid of 0.5° × 500 m (AAO × altitude), which corresponds to 56 km × 500 m in the horizontal and vertical direction. In the following we refer to this OSIRIS tomographic retrieval grid as OSIRIS "pixel" to be consistent with the terminology used for CIPS. The actual horizontal resolution of the retrieval is coarser than this grid and is defined by the retrieval's averaging kernel. To retrieve cloud information at a given point, the tomographic algorithm combines information from a large number of nearby lines of sight, each covering several hundred kilometers of path through the PMC layer. In order to characterize the ability of the algorithm to resolve cloud structures, measurement simulations have been performed on a cloud filling a single OSIRIS pixel. These result in averaging kernels that are typically described by a Gaussian shape with a FWHM of 280 km. In the case of the OSIRIS tomography, the resolution is limited by the relatively sparse distribution of measurements. Limb radiances from individual lines of sight are recorded once per second, while the satellite scans the tangent altitudes with a speed of about 0.75 km s$^{-1}$, and moves along its orbit at about 7 km s$^{-1}$. As a result, the ability to resolve a given cloud structure varies stochastically depending on how the nearest lines of sight happen to be distributed relative to the cloud structure. Applying an averaging kernel of 280 km to the data will thus account for lowest possible resolution, although if a cloud structure is in a more favourable position to the closest lines of sights, the ability to resolve this structure increases. To account for this, horizontal resolutions of both 280 km (typical averaging kernel) and 56 km (length of OSIRIS retrieval pixel) will be considered as limiting cases when comparing individual OSIRIS and CIPS measurements in Section 4.2.

The measured PMC limb radiance contains contributions from molecular Rayleigh scattering from the background atmosphere as well as instrumental effects, e.g., baffle scattering and offset due to dark current. Therefore, a separation of the pure cloud signal from the background is needed before the tomographic retrieval can be performed. As the short limb scans in tomographic mode do not cover tangent altitudes outside the PMC regions, these background signals cannot be measured independently. Rather, the molecular scattering background is estimated by calculations of Rayleigh scattering based on an atmospheric density profile taken from MSIS (Hedin, 1991). The contribution to the signal from the instrumental effects is calculated as the mean value of the background taken during the ordinary limb scans measured the days before and after the tomographic scans.

Spectral analysis of OSIRIS PMC data enables the retrieval of cloud microphysical properties such as mode radius, number density and IMD (Karlsson and Gumbel, 2005). Here we provide a brief description of the spectral analysis on tomographic data. For a detailed description, together with a discussion about uncertainties, the reader is referred to Hultgren and Gumbel (2014). The spectral analysis of the microphysical properties mode radius $r_m$[nm], particle number density N [cm$^{-3}$] and IMD IMD [ng m$^{-3}$] is based on the local scattering coefficient $\beta_\lambda$[m$^{-1}$ sr$^{-1}$] from the tomographic retrieval in seven different wavelength bands between 277.3 nm and 304.3 (see Table 1 in Karlsson and Gumbel, 2005). The local scattering coefficient is related to the local PMC particle population by Eq. (1)

$$\beta(\lambda) = N \int f(r, r_m) \frac{\partial \sigma}{\partial \Omega} (r, \lambda) dr ,  \tag{1}$$

where $f(r, r_m)$ is the normalized particle size distribution and $\partial \sigma / \partial \Omega$ is the differential scattering cross section for the direction in question. It is calculated using theoretical scattering spectra from numerical T-matrix simulations (Mishchenko and Travis, 1998; Baumgarten and Fiedler, 2008). The scattering cross section is dependent on the material of the particles as well as their shape and size, so assumptions of these properties are needed for retrieving the mode radius. The particles are assumed to consist of water ice, have the shape of oblate spheroids with an axial ratio of 2. The particle size distribution $f(r, r_m)$ is assumed to be a normal distribution with a width that varies depending on mode radius as $\Delta r \approx 0.39 \times r_m$ up to $r_m = 40$ nm and are constant at 15.8 nm for larger particles. These assumptions are based on the findings of previous studies

(Hervig et al., 2009; Baumgarten et al., 2010) and are the same as the assumptions used in the CIPS PMC retrievals (Lumpe et al., 2013). $\beta$ is fitted using an Angström exponent, i.e., assuming an exponential dependence on wavelength (von Savigny et al, 2004). Given the above assumptions, comparing the observed spectral dependence to theoretical calculations yields the mode radius. Once $\beta(\lambda)$ and $r_m$ have been retrieved and $\partial\sigma/\partial\Omega$ has been calculated with the T-matrix simulations, N can be determined from equation (1). Finally, combining the number density, size distribution, and particle shapes yields the total ice volume and IMD (Hultgren and Gumbel 2014).

### 2.1.1 Discussion of OSIRIS tomography uncertainties

It is important to assess the uncertainty of OSIRIS tomography PMC data products. Here we refine the uncertainty discussion provided by Hultgren and Gumbel (2014). The starting point is the uncertainty of the limb radiances measured by OSIRIS. These uncertainties propagate through the tomographic retrieval of the local scattering coefficients, and then through the subsequent spectral analysis of the microphysical cloud properties. As for the accuracy of the OSIRIS limb radiances, there is an absolute error of about 10%, which is a combination of the uncertainties due to instrument calibration and due to the influence of ozone absorption on the limb radiances in the ultraviolet (Benze et al., 2018). As for the estimated random error of the OSIRIS limb radiances, Hultgren and Gumbel (2014) list as error sources the instrument noise as well as uncertainties in the subtraction of the molecular and instrumental background. For faint PMCs, the random error is directly related to the need to discriminate the cloud signal from the molecular background (and its random error). Another random error is introduced by mesospheric ozone: Absorption by ozone along the LOS has a significant effect on limb measurements of PMCs in the ultraviolet. This is a particular concern for tomographic retrievals as these include lines of sight with tangent altitudes well below 80 km where ozone absorption increases substantially. Lacking direct ozone measurements in conjunction with the PMC measurements, we treat the natural variability of ozone in the upper mesosphere as a contribution to the random error of the PMC limb radiance.

The propagation of these limb radiance uncertainties through the tomographic retrieval is investigated by a Monte Carlo approach, i.e., by running the retrieval with limb radiances randomly perturbed within the uncertainty limits (Hultgren and Gumbel, 2014). The resulting relative error of the PMC volume scattering coefficient $\beta$ approaches 100% at the PMC detection limit ($\beta \approx 10^{-10}$ m$^{-1}$ sr$^{-1}$), and is about 10% for cloud volumes with $\beta \approx 10^{-9}$ m$^{-1}$ sr$^{-1}$, which corresponds to a typical cloud brightness above which the OSIRIS spectroscopic size retrieval is meaningful. For the brightest PMC volumes, the estimated random error is a few percent, mainly determined by the influence of ozone absorption. When comparing the OSIRIS tomographic results to PMC measurements with a higher spatial resolution like CIPS or lidar, it is important to note that it is not necessarily the above errors that limit the comparison. As stated by Hultgren et al. (2013), measurement and retrieval errors are sufficiently small not to limit the sharpness of retrieved PMC structures. Rather, knowledge about local PMC properties is limited by the retrieval resolution of the OSIRIS tomography, i.e., the averaging kernels.

As described above, the tomographic retrieval of the scattering coefficient at several wavelengths is the basis for the subsequent spectral analysis of PMC microphysical properties like mode radius, number density or IMD. Hultgren and Gumbel (2014) have discussed the corresponding error propagation. In particular they point out that the IMD is rather independent of the uncertainties in the retrieval of particle size and particle number density, and rather independent of the assumptions on the size distribution. This is due to the fact that the radius dependence of particle volume and of particle scattering cross section to a large extent cancel each other when inferring the ice mass. As a result, the local PMC IMD is to a large extent, albeit not completely, proportional to the local PMC scattering coefficient. The uncertainty of IMD is subject to the same absolute error of ~10% as the scattering coefficient (due to instrument calibration and influence of ozone absorption on the limb radiances). Additionally, propagating the error in local scattering coefficient in the seven different wavelength regions through the spectral

analysis provides the random error of OSIRIS IMD. In section 4, the PMC properties inferred from OSIRIS will be compared to CIPS. To this end, the local cloud properties from the tomographic retrieval will be vertically integrated to cloud column properties like directional albedo or IWC. Accordingly, the above errors of the local PMC properties will be propagated to an error of the vertical column properties.

**Table 1. Summary of OSIRIS PMC uncertainties**

| Parameter | Description | Accuracy |
|---|---|---|
| PMC volume scatter coefficient $\beta$ | Uncertainty estimated by Monte Carlo approach based on limb radiance uncertainty | ~10% |
| PMC Ice Mass Density | Uncertainty estimated by propagating uncertainty of $\beta$ in 7 different wavelength regions through spectral analysis | ~10% |

Given the above uncertainties, it is of importance to assess the accuracy of tomographic OSIRIS cloud retrievals by comparison with cloud properties derived from independent measurements and model simulations. As for the latter, Megner et al. (2016) compared the OSIRIS tomographic retrieval of cloud properties to the Community Aerosol and Radiation Model for Atmospheres (CARMA) (Toon et al., 1979; Turco et al., 1979). They investigated which part of the ice particle size distribution OSIRIS captures and evaluated how this affects the retrieved cloud properties. They concluded that the OSIRIS tomographic IWC is within approximately 20% of the simulated IWC, however that mean radius and number density are less accurate. Specifically, the tomographic retrieval performs well for retrieving mode radius in the range 50-70 nm, but overestimates mode radius (up to a factor 3) for small mode radii and underestimates it for large mode radii (80 nm and above). Moreover, the study by Megner et al. (2016) suggested that the tomographic algorithm performs well for retrieving number density for small number densities (which usually occur lower in the cloud) but greatly underestimates it for high number densities (which usually occur higher up in the clouds, where the retrieval misses the smaller mode radius).

## 2.2 AIM CIPS

The AIM satellite, launched in 2007, was the first satellite mission fully dedicated to study PMCs, with an overall goal to resolve how PMCs form and vary (Russell et al., 2009). AIM is moving in a circular, sun-synchronous orbit near 600 km altitude. AIM currently has two operating instruments, the Solar Occultation for Ice Experiment (SOFIE) (Gordley et al., 2009) and the high-resolution UV panoramic imager CIPS. CIPS consists of four nadir-pointing wide-angle cameras with a combined FOV of 120° by 80° and operates in the UV with a spectral band centered at 265 nm with a width of 15 nm (McClintock et al., 2009). In contrast to the majority of the previous satellites and in-situ measurements of PMCs that apply the wavelength-dependence of the scattering or extinction derive cloud properties, CIPS utilizes the angular dependence of PMC scattering. By observing the clouds and the background atmosphere from a range of scattering angles, the clouds can be separated from the bright Rayleigh scattering background by taking advantage of the fact that the anisotropy of light scattering depends on the size of the scatterer. For background subtraction, the detection algorithm uses the well-established assumption that a PMC particle is a strong forward-scatter and highly asymmetric, whereas the background Rayleigh signal is symmetric about 90 deg. When the cloud particle scattering phase function has been distinguished from the Rayleigh background, the particle radius can be characterized by matching the retrieved phase function to a set of phase functions derived from T-matrix calculations (Bailey et al., 2009).

### 2.2.1 Discussion of CIPS PMC uncertainties

The CIPS retrieval algorithms and data products with error analysis and cloud detection sensitivity have been described in detail by Lumpe et al. (2013). For this study, we utilize CIPS level 2 data, which is the primary CIPS PMC data product

consisting of cloud presence, cloud albedo normalized to 90° scattering angle and 0° view angle [$10^{-6}$ sr$^{-1}$], mean particle radius [nm], column ice reported as ice water content [g km$^{-2}$] and ice column density [cm$^{-2}$], derived phase function and geolocation. We use CIPS data version 4.20, which has a horizontal resolution of 25 km$^2$ at latitudes between 55° and 84°(Lumpe et al., 2013). CIPS cloud detections and albedo values were previously compared to the solar backscatter ultraviolet (SBUV/2) instruments (Benze et al., 2009; 2011). This was accomplished by applying a ''SBUV-type'' algorithm to the CIPS level 1A data to make the two datasets comparable. Cloud frequency and brightness from CIPS were shown to be in good agreement with SBUV/2 retrievals. By downgrading CIPS data to the SBUV resolution, it was shown that CIPS cloud frequency and albedo were in good agreement with the SBUV data. CIPS albedo has a bias (estimated albedo uncertainty) that varies depending on solar zenith angle (SZA), radius and albedo (Fig. 21 in Lumpe et al., 2013). For the range of SZA used in this study (59° to 71°), the albedo is biased low by ~0.5e-6 sr$^{-1}$. In this study, we correct for this in every CIPS pixel by manually adding 0.5e-6 sr$^{-1}$ to all finite non-zero pixels, in line with Benze et al. (2018). However, it should be noted that this correction is not done in the operational CIPS product.

The data contains quality flags that indicate the number of scattering angles that were used to determine the scattering phase function. An observation is most robust when at least 6 scattering angles are used (marked with quality flag 0), and more uncertain when fewer angles are used (marked with quality flag 1 for 4 or 5 angles and quality flag 2 for fewer than 4 angles). In line with the recommendation for satellite comparisons of microphysical properties, we use data flagged with 0 or 1 in this work.

The CIPS retrieval algorithm and error analysis have been outlined and discussed by Lumpe et al. (2013). The random albedo uncertainty in each CIPS pixel is given as 1e-6 sr$^{-1}$. Caution is warranted if investigating albedo or IWC for cloud pixels with albedo < 1-2e-6 sr$^{-1}$, but the results might still be valid. Specifically, for qualitative comparison of horizontal structures in albedo and retrieved cloud parameters, the recommendation from the CIPS team is to use a detection threshold of 1e-6 sr$^{-1}$ to screen out false detections, and for quantitative comparisons of albedo and retrieved cloud parameters, a detection threshold of 2e-6 sr$^{-1}$ should be applied. In the level 2 data, all dim pixels with albedo < 1e-6 sr$^{-1}$ are set to NaN by the CIPS team. Following the above ideas, for the quantitative comparisons of albedo, we apply a detection threshold of 2e-6 sr$^{-1}$, and choose to manually set to zero any pixels with a retrieved albedo of less than 2e-6 sr$^{-1}$. The uncertainty connected to the handling of the dim CIPS pixels will be discussed in Sect. 4.1. On the other hand, for qualitative comparisons of horizontal structures along individual orbits, we use a detection threshold of 1e-6 sr$^{-1}$. We note that for CV regions that contain many CIPS pixels with albedo less than 2e-6 sr$^{-1}$, setting all of these pixels to NaN could result in a positive bias of CIPS compared to OSIRIS, since only the higher-albedo CIPS pixels would be included in the analysis. In addition, we restrict our albedo comparison to CIPS pixels that have a valid retrieved mode radius of > 20 nm, since we need to correct CIPS albedo due to the differences in scattering conditions, and this transformation is dependent on mode radius (described in detail in section 3.3).

CIPS IWC is completely determined by albedo and radius. As discussed in Lumpe et al. (2013), the information content of scattering angle variation of the albedo observed by CIPS decreases for decreasing particle sizes. By comparing PMC retrievals to simulated data (their Fig. 21), systematic error and random error were estimated. For IWC, it was shown that the systematic error is highly dependent on mode radius. For large particles (> 40 nm), the systematic error shows only little dependence on IWC itself or on mode radius, and is generally < 10 g km$^{-2}$. For smaller particles (between 20-40 nm), and especially for low SZA, the systematic error is estimated to be -20 to -40 g km$^{-2}$. For the range of SZA used in the current study and for mode radius of 30 nm, the systematic error is -10 g km$^{-2}$ for IWC of 100 g km$^{-2}$ and -20 g km$^{-2}$ for IWC of 250 g km$^{-2}$. More recently, Bailey et al. (2015) compared CIPS IWC to SOFIE IWC by a common volume comparison of data from 2007-2009 and noted that for mean particle radii < 20 nm most PMCs are not detectable for CIPS, although for mean particle radii > 30 nm, CIPS is able to detect most PMCs. In the current study, we limit our comparison of IWC between CIPS and OSIRIS tomography to

CIPS pixels where a mode radius > 20 nm is retrieved. To be comparable to OSIRIS, the cloud properties in each pixel are horizontally averaged over the CIPS CV, and to be consistent with the way OSIRIS errors are treated above, CIPS pixel-by-pixel albedo errors and IWC errors are propagated to a horizontally averaged error for the respective properties in the CV.

## 3 Method

In the present study, we use the Odin OSIRIS dataset from two northern hemispheric seasons, 2010 and 2011. During these seasons, the tomographic orbits were scheduled to coincide in both time and space with CIPS. To date, Odin has continued to be run in the tomographic mode for specific orbits every PMC season for both the northern hemisphere and the southern hemisphere. The 2010-2011 dataset consists of a total of 180 orbits that have been performed over 12 days in the two mentioned northern hemispheric PMC seasons. In 2010, the Odin/OSIRIS tomographic scans covered tangent altitudes around 74-88 km; in 2011 this range was adjusted to about 76-87 km.

### 3.1 Coincidence criteria

Temporal and spatial coincidence criteria can vary broadly for validation studies of satellite instruments depending on measuring technique and comparison quantity. The fact that PMCs are small-scale and variable phenomena places high demands on the spatial and temporal coincidence criteria. The time period for this study extends from June to August in 2010 and 2011 (see also Table 1 in Hultgren et al., 2013). CIPS – OSIRIS coincidences occur between 78°N and 80°N where the orbits of both satellites cross and produce common volume PMC observations at ~15.45 local time. We have chosen a criterion to ensure a sufficient number of observations to make a statistically significant comparison and to minimize the error due to the time difference between coincident measurements. The time it takes the OSIRIS instrument to obtain a sufficient number of lines of sight through a given cloud volume for the tomographic retrieval is about 1 minute. As CIPS needs to take observations under different scattering angles to perform its phase function analysis, it takes about 6 minutes to carry out the necessary observations of the cloud common volume. In this study, we chose a time-constraint of 5 minutes as the temporal coincidence criteria to prevent the time constraint from causing a larger error in the comparison than what is introduced by the CIPS retrieval itself. To test the sensitivity of the comparison results to our requirements in time, we performed comparisons with shorter time constraints. These analyses did not show any systematic difference due to the time difference between the observations within the range of 5 minutes. The spatial coincidence is limited by the width of the CIPS pixels, which is 5 km in the nadir.

### 3.2 CIPS and OSIRIS Common Volume

The common volume observations occur between 78° N and 80° N. In this latitude band, the satellite orbits of OSIRIS and CIPS coincide both in space and in time during 14-15 overpasses each day when the satellites are in ascending node. As the orbit periods are slightly different, ~96 minutes for Odin and ~95 minutes for AIM, these coincidences occur approximately every 30 days. The tomographic orbits were scheduled to be performed so that common volume observations with CIPS would be possible. We define the CIPS/OSIRIS CV as the observations in the CIPS measurement footprint that overlap the OSIRIS tomographic observations. The height of the CV is defined by the vertical extent of the tomographic data, which ranges from 76 to 90 km. Furthermore, we introduce a CIPS/OSIRIS CV "element" as a subset of the larger CIPS/OSIRIS CV. Figure 1 provides an example of a CIPS/OSIRIS coincidence in which CIPS albedo is plotted on a lambert conformal conic projection. The white line indicates those ~660 CIPS pixels that are included in the CV, and the red square indicates the horizontal extent of those 66 CIPS pixels that are included in a single CV element. Each CV is comprised of ~10 CV elements. From an OSIRIS perspective, the length of each CV element is defined by the length of one OSIRIS tomographic retrieval pixel, which is 56 km in the direction along the satellite orbit. The width of each CV element is defined by the OSIRIS FOV at the tangent point,

perpendicular to the LOS, which is 30 km. Thus each CV element measures 56 km along track by 30 km cross-track. From a CIPS perspective, the CV element is defined as the set of CIPS pixels that overlap the horizontal extent of one OSIRIS tomographic retrieval pixel. Since v4.20 CIPS level 2 data has a resolution of 5x5 km in the nadir, the overlapping region contains ~66 CIPS pixels. Note that even though the CV element is fairly limited in horizontal extent, it is evident from Fig. 1 that the cloud albedo therein can be quite variable. Figure 2 shows the distribution of CIPS CV elements that are filled with cloudy pixels. The total number of CV elements used to produce this figure is 1513. The CIPS CV element is completely filled with cloudy pixels for a large part of the coincidences (>500), but for ~300 coincidences, the CV element is not containing any cloud data. Figure 2 shows an apparent dominance of CV that are either almost cloud-free (0%) or cloud-filled (100%). This is related to the choice of the size of the CV. If a larger CV would have been used, the distribution would not show such high numbers of detections for 0 and 100% clouds fraction. The CIPS CV element consisting of 66 CIPS retrieval pixels with an area of 25 $km^2$ each has a total horizontal extent of 1650 $km^2$, while the horizontal extent of each OSIRIS CV element is 30 km × 56 km = 1680 $km^2$. The difference in horizontal extent between one CIPS CV element and one OSIRIS CV element is smaller than 2% and is considered negligible in our study.

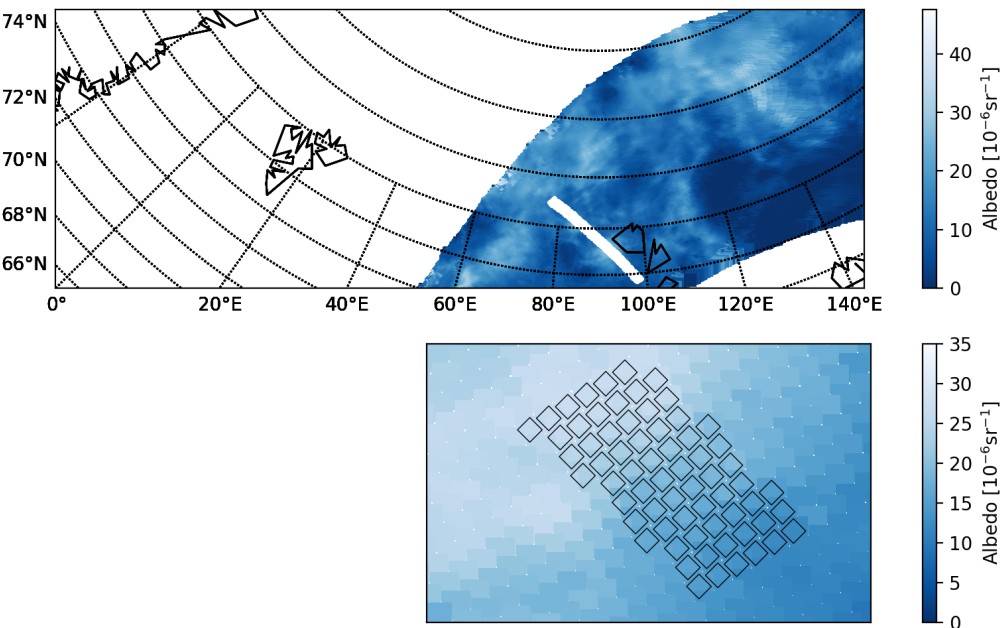

**Figure 1:** Example orbit showing a CIPS/OSIRIS coincidence on a polar map plot for CIPS orbit 50777 and OSIRIS orbit 17098. The top panel shows the CIPS orbit strip. The white line on top of the CIPS orbit strip indicates the overlapping ~660 CIPS pixels in the CV. Each CV is composed of ~10 CV elements. The bottom panel shows an example of only those 66 pixels contained within one CV element.

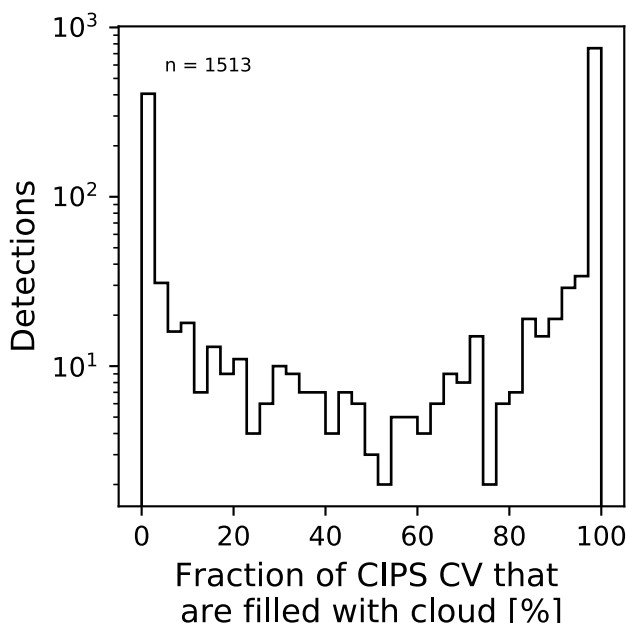

**Figure 2: AIM CIPS** occurrence of PMCs in the common volume with OSIRIS during the observations in 2010-2011. The plot shows the number of CVs containing a certain fraction of CIPS pixels with identified PMCs. Average latitude is 80° N, local time ~15.45. The meancloud fraction is 62%.

### 3.3 Method to make limb and nadir observations comparable in the common volume

Recently, Benze et al. (2018) demonstrated a method for comparing PMC observations from the normal (non-tomographic) OSIRIS limb scans to CIPS. This method took into account the measurement geometry and instrument sensitivity. By performing a detailed common volume comparison, their study showed that the PMC brightness from the normal OSIRIS scans agrees well on average (-9 ± 14%) with the CIPS brightness. The method for the PMC satellite comparison described in their study provides the basis for the comparison method in the current study. However, some important modifications are necessary for the more detailed tomography comparison.

**Table 2. Overview of parameters**

| Dataset | Data product used for inferring cloud brightness | Data product used for inferring cloud ice | Retrieval grid resolution |
|---|---|---|---|
| CIPS v4.20 | Cloud albedo, $A\ [sr^{-1}]$ | Ice mass density, $IMD\ [ng\ m^{-3}]$ | 5 km x 5 km in nadir (horizontal plane) |
| OSIRIS tomography v1.6 | Local Scattering Coeff., $\beta_\lambda\ [m^{-1}sr^{-1}]$ | Ice water content, $IWC\ [\mu g\ m^{-2}]$ | 56 km x 500 m (vertical plane) |

When comparing observations from two instruments with different viewing geometry and resolution it is necessary to both define the appropriate common volume and make the observational quantities comparable. The signal from each satellite needs to be integrated to fill the common volume. The primary PMC product of OSIRIS is volume scattering coefficient $[m^{-1}\ sr^{-1}]$, while the primary PMC product of CIPS is directional albedo at 90° scattering angle $[sr^{-1}]$. The OSIRIS volume scattering coefficient has a coarse horizontal resolution, but a high vertical resolution, while the CIPS albedo has a high horizontal resolution but no vertical resolution (Table 2). To make these two quantities of PMC cloud brightness comparable, one needs to define a common quantity in the common volume that accounts for the difference in vertical and horizontal resolution.

Since the OSIRIS tomographic PMC products are reported on a vertical-horizontal plane with the vertical axis as altitude (rather than tangent altitude as for the normal OSIRIS PMC retrievals), $\beta_\lambda$ can be converted into albedo (A) in units of sr$^{-1}$ by integrating over the vertical column, assuming optically thin clouds. To clarify, we integrate the OSIRIS scattering coefficient vertically to answer the question: What albedo would CIPS retrieve from the same PMC volume? The resulting albedo from OSIRIS can be expressed as Eq. (2):

$$A_{OSIRIS} = \int_{76km}^{90km} \beta_{277nm} \, dz \,, \tag{2}$$

where the vertical integration limits of 76 km and 90 km are given by the vertical extent of the tomographic dataset, and $\beta_{277nm}$ is the retrieved volume scattering coefficient at the wavelength 277 nm.

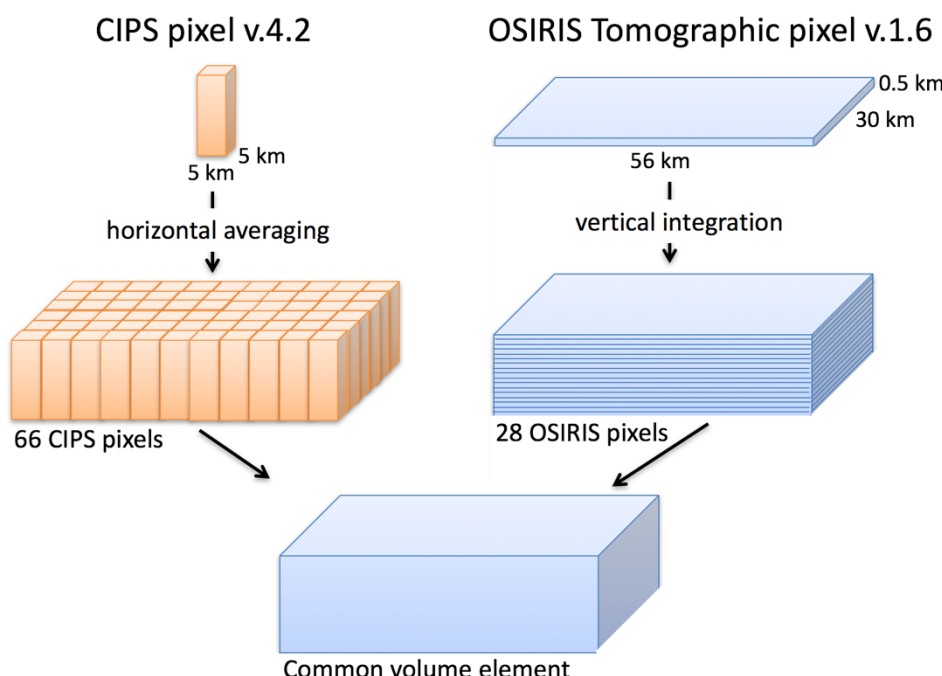

**Figure 3**: The 3D common volume is defined by the horizontal extent of one OSIRIS tomographic pixel (56km*30km) and the vertical range (76 to 90 km) of the OSIRIS tomography observations. CIPS pixels have a size of 25km$^2$ with each side 5 km in the nadir. In order to make CIPS comparable to OSIRIS in the CV, 66 CIPS pixels (11*6) are averaged over the CV. The tomographic OSIRIS pixels need to be integrated over the column of 28 levels to be comparable to CIPS.

Albedo at a scattering angle of 90° for CIPS in the common volume element (x, y) is defined as Eq. (3):

$$A_{CIPS} = \text{mean}\left(A_{265nm}^{90°}(x, y)\right), \tag{3}$$

where 90° refers to the scattering angle that CIPS albedo observations are normalized to, and 265 nm refers to the region of the spectrum where CIPS operates.

The two quantities, $A_{OSIRIS}$ and $A_{CIPS}$, cannot be compared directly since the satellites observe the PMCs in different parts of the UV range and observe the clouds under different scattering conditions. The next section describes how this is handled to make the observations comparable in the common volume elements.

As mentioned earlier, OSIRIS observes PMCs and the background atmosphere at wavelengths between 275 and 810 nm with a resolution of about 1 nm, while CIPS observes PMCs in a wavelength region centered at 265 nm with a width of 15 nm. The tomographic retrieval provides the local scattering coefficient for seven wavelength intervals in the UV (centered at 277.3, 283.5, 287.8, 291.2, 294.4, 300.2 and 304.3 nm, also see Table 1 in Karlsson and Gumbel, 2005) and uses these to retrieve the microphysical properties. The reported CIPS albedo is normalized to a solar scattering angle of 90°, while the OSIRIS local

scattering coefficient in the overlapping volume is measured at solar scattering angles between 75° and 78°. To make the two quantities comparable, a correction for the differences in scattering conditions and wavelengths is necessary. In this study, we convert CIPS albedo into the conditions used for OSIRIS. The conversion factors depend on the PMC scattering properties and thus on the size and shape of the PMC particles. In this study, we base our conversion factors on the particle sizes retrieved from CIPS, using the same assumptions as in the operational CIPS retrievals, namely that the particles consist of water ice, that the particle shape is oblate spheroids with axial ratio 2, and that the particle size distribution is a normal distribution with width varying as a function of mode radius as $\Delta r \approx 0.39 \times r_m$ up to $r_m = 40$ nm and approximately constant above (Baumgarten et al., 2010) (Lumpe et al., 2013). The conversion factors are obtained from numerical T-matrix simulations (Mishchenko and Travis, 1998). The spectral conversion factor $C_{phase}$ range between 0.39-5.57 for particles between 1-100 nm. $C_{phase}$ increase with increasing particle size. For particles in the range 1-20 nm $C_{phase}$ varies between 1.0-1.5, for 21-50 nm $C_{phase}$ varies between 0.6-2.7 and for particles in the range 51-100 nm $C_{phase}$ varies between 0.4-5.6. The spectral conversion factor $C_{spectral}$ range between 0.8 to 1.0.

Each CIPS pixel is transformed into scattering conditions of OSIRIS by multiplication of the conversion factors $C_{spectral}$ and $C_{phase}$ corresponding to the retrieved CIPS radius in the same pixel and subsequently averaged over the CV element to generate a mean albedo comparable to OSIRIS. The mean CIPS albedo in the CIPS CV element can be expressed as Eq. (4):

$$A_{CIPS}^{transformed\ CV\ element} = mean\left(C_{spectral(90° \text{ to OSIRIS SSA}°)} \cdot C_{phase(265\ nm\ to\ 277.4\ nm)} \cdot A_{265nm}^{90°}(x,y)\right) \tag{4}$$

When comparing OSIRIS and CIPS data, these conversion factors introduce an additional uncertainty, beyond the specific uncertainties of the OSIRIS and CIPS data sets. These uncertainties will be discussed in Section 4.1. As noted in section 2.2.1, we manually set all dim CIPS pixels < 2e-6 sr$^{-1}$ to zero in the qualitative comparison. These zero pixels have been included in the average of CIPS mean albedo.

Besides comparing the cloud albedo between the instruments, we also extend our study by comparing the cloud ice. While OSIRIS tomography reports cloud ice as ice mass density (IMD)in units of [ng m$^{-3}$], CIPS reports the column integrated quantity ice water content in units of [$\mu g$ m$^{-2}$]. To make these two quantities comparable in the common volume element, we adopt the same integration method as for comparing albedo described in the section above. The OSIRIS IWC can be calculated from a vertical integration of OSIRIS IMD following Eq. (5):

$$IWC_{OSIRIS}^{CV\ element} = \int_{76km}^{90km} IMD\ dz \tag{5}$$

The appropriate CIPS value of IWC is the mean value of IWC in all pixels in the CV element, including all the ice-free pixels following Eq. (6):

$$IWC_{CIPS}^{CV\ element} = mean\left(IWC\ (x,y)\right) \tag{6}$$

The microphysical properties IWC$_{OSIRIS}$ and IWC$_{CIPS}$ are directly comparable - in contrast to albedo, no transformation factors are necessary for the comparison.

### 3.4 Subset selection

The statistical analysis of albedo is based on 180 coinciding satellite orbits from NH 2010 and NH 2011. Out of these 180 orbits, 2 OSIRIS orbits from NH 2010 had to be discarded due to geolocation errors. For each common volume observation, approximately 14 CV element observations with CIPS are performed making a total of approximately 2492 possible CV element observations available for the statistical analysis for the 2010 and 2011 seasons. The coarser horizontal resolution of OSIRIS requires the application of appropriate averaging to the CIPS data before a comparison of retrieved cloud properties is feasible. As discussed in Section 2.1, the OSIRIS averaging kernel can be represented by a Gaussian distribution with

FWHM of 280 km. As a consequence, each comparison between CIPS and OSIRIS requires averaging the CIPS data over several of the common volume elements of size 55 × 30 km. On the edges of the CIPS orbit strip, this demand puts restrictions on the number of CIPS common volume elements that can be included in the comparison, reducing the available number of common volume elements per OSIRIS orbit from 14 to approximately 10 and the total number of common volume elements from 2492 to ~1500. Limiting the statistical analysis to common volume elements containing only good quality pixels based on CIPS quality flags (discussed in Section 2.2.1) further reduces the number of common volumes to 1292.

## 4 Results

We have performed a statistical analysis of albedo and IWC between OSIRIS and CIPS in the common volume. To gain deeper insight into the relationship between both datasets, we have also compared albedo and IWC along single orbits. The previous section described the coincidence criteria and the method for making the PMC data from OSIRIS and CIPS comparable in the common volume. The cloud properties from each instrument are made comparable in the common volume by integrating OSIRIS local scattering coefficient and IMD vertically and taking the horizontal mean of the CIPS albedo and IWC. In addition, as described in the previous section, the CIPS albedo is transformed into the scattering conditions of OSIRIS. A coincidence criterion of 5 minutes has been applied, and a preliminary selection based on the quality flag has been performed to eliminate questionable data pixels. This section is divided into two different parts, starting with the statistical comparison of albedo and IWC and the uncertainties related to the choice of comparison method used in this study in Sect 4.1. In Sect 4.2 we present the results from the individual orbits.

### 4.1 Quantitative comparison of OSIRIS albedo with CIPS

Figure 4 shows a scatter plot comparing the OSIRIS albedo and CIPS albedo in the common volume for the total set of 1292 observations. Each dot represents the albedo inferred from OSIRIS and the corresponding albedo inferred from CIPS in the common volume element, and the grey dashed line denotes the one-to-one line. Figure 4 shows that OSIRIS albedo and CIPS albedo generally agree well both for faint clouds and bright clouds, although for most cases, OSIRIS albedo is higher than CIPS. We determine the offset between the OSIRIS and CIPS albedo results as 2.8e-6 sr$^{-1}$ (±2.4e-6 sr$^{-1}$). The uncertainty of this offset has been estimated based on the errors of the individual OSIRIS and CIPS albedos in figure 4. As a limb viewing instrument, OSIRIS observes PMCs as enhancements in the limb radiance and integrates the signal over a long distance, while CIPS as a nadir instrument observes PMCs as enhancements in the brighter Rayleigh background and integrates the signal over a much shorter distance. OSIRIS can, therefore, observe fainter clouds than CIPS. To quantify the agreement between the instruments for various cloud regions, we sort the observations into three regions based on OSIRIS albedo: faint clouds (0-10e-6 sr$^{-1}$), medium clouds (10-30e-6 sr$^{-1}$) and bright clouds (30-80e-6 sr$^{-1}$) and calculate the Pearsson correlation coefficient for each region. For faint clouds, the correlation is 0.81, for middle bright clouds the correlation is 0.85 and for bright clouds, the correlation is 0.92. The correlation coefficient for the total set of observation is calculated to be 0.97. The small systematic bias of 2.8e-6 sr$^{-1}$ (±2.4e-6 sr$^{-1}$) between CIPS and OSIRIS together with an overall correlation coefficient of 0.97 for the 1292 number of CV elements shows a very good agreement between the instruments and analysis methods.

The error bars in Fig. 4 represent the error from the basic uncertainty discussed in detail in Sect. 2.1.1 (OSIRIS) and Sect. 2.2.1 (CIPS), combined with the error that is introduced by transforming the data into a comparable property in the CV. For OSIRIS, the directional albedo is, as previously mentioned, obtained as the vertical column integral over the scattering coefficients. In order to represent the uncertainty of the column albedo caused by this integration, the error bars of the albedo include the root of the sum of the squares of both the systematic error (10% due to calibration) and the estimated random error of the scattering coefficient described in Section 2. The resulting error of the OSIRIS albedo is dominated by the uncertainty of the brightest

PMC pixels in the column. For dim or PMC-free areas, the albedo error is essentially determined by the PMC retrieval threshold at each altitude, which sums up vertically to about 0.2e-6 sr$^{-1}$. This number can be interpreted as a PMC detection threshold of OSIRIS expressed in terms of directional albedo.

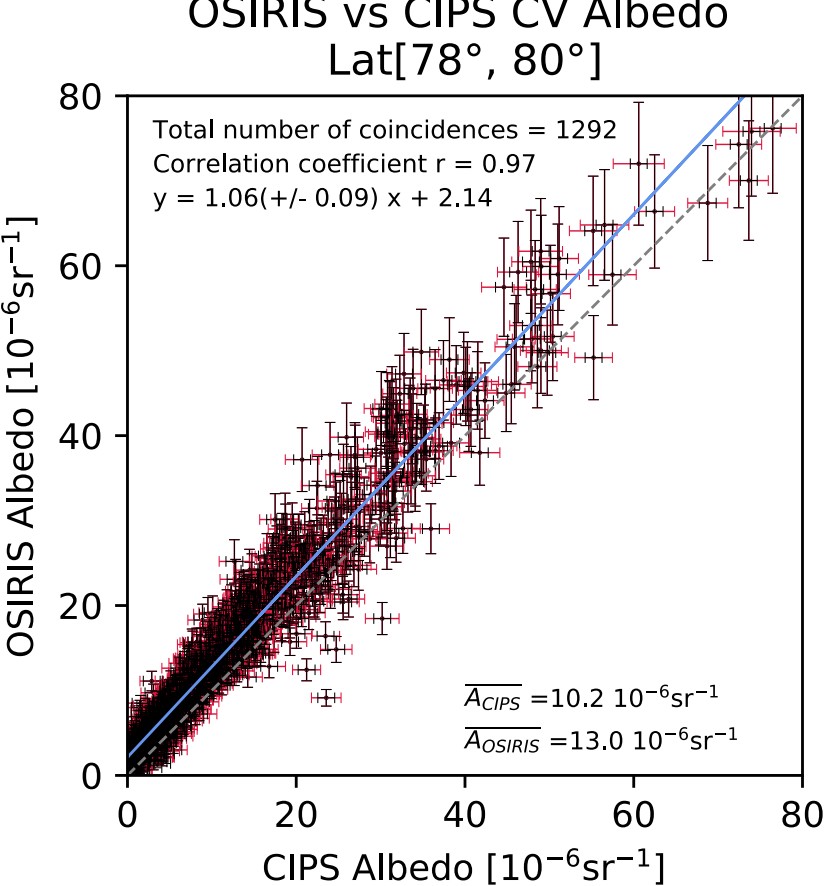

**Figure 4**: Scatterplot of OSIRIS and CIPS observations of albedo in the common volume. The grey dashed line denotes the one-to-one line. The blue line is the regression line. The average Albedo for CIPS and OSIRIS for all the CV in the figure is indicated in the bottom right of the figure. OSIRIS error bars are a combination of systematic and statistical uncertainty. CIPS error bars are a combination of statistical uncertainty and uncertainty due to handling of dim cloud pixels (black), while the extended error bars (red) denote the uncertainty that is introduced by conversion factors that accounts in the difference in wavelength and scattering angle between the instruments. The reader is referred to the error discussion in the text for a more detailed description of CIPS and OSIRIS errors.

For CIPS, the directional albedo is obtained as the horizontal average of the pixels contained in the CV element. The corresponding uncertainty is a combination of the uncertainty due to the conversion factors, a statistical uncertainty (of 1e-6 sr$^{-1}$), and an uncertainty due to the handling of dim cloud pixels. The following text describes how these CIPS uncertainties are handled in our study. As described in Section 3.3, conversion factors must be applied in order to make the observation conditions of OSRIS and CIPS comparable. $C_{spectral}$ accounts for the difference in wavelengths, $C_{phase}$ accounts for the difference in scattering angle. These factors depend on the particle size and are calculated based on the CIPS size retrievals. They introduce an additional uncertainty to the OSIRIS/CIPS albedo comparison, beyond the specific uncertainties of both datasets. This uncertainty can be assessed based on the uncertainty of the CIPS particle size retrieval. Lumpe et al. (2013, Fig. 21) discuss the CIPS radius uncertainty as a function of PMC albedo and solar zenith angle. We thus obtain an uncertainty of the conversion factors by applying these radius uncertainty ranges when calculating the conversion factors from our T-matrix approach. The resulting uncertainties of the conversion factors are included as additional error bars on the CIPS data, marked in red in Figs. 4 and 6. Note that these red error bars do not denote uncertainties inherent to CIPS but uncertainties of our OSIRIS/CIPS comparison method. In order to represent the uncertainty of the mean CIPS albedo caused by the horizontal averaging over all the CIPS pixels in the common volume element, it is necessary to consider the handling of CIPS dim pixels in the current study. By manually setting all CIPS pixels where albedo < 2e-6 sr$^{-1}$ to 0, an additional statistical uncertainty is

introduced to the mean CIPS albedo. This uncertainty can be easily tested by setting the albedo in these CIPS pixels to either 0 (lower limit) or 2e-6 sr$^{-1}$ (upper limit). It turns out that the resulting uncertainty range of the mean CIPS albedo varies linearly with albedo for the dim clouds. A linear fit then provides an expression of this "dim cloud" uncertainty as a function of albedo. In the albedo range of 0 - 7.5e-6 sr$^{-1}$, the resulting error due to "dim clouds" can be represented as -0.2 x CIPS mean albedo

+2.5, and 0 for CIPS mean albedo > 7.5e-6 sr$^{-1}$. -0.2 is the slope and +2.5 is the intercept. The black error bars in Figs. 4 and 6 represent the square-sum of the statistical uncertainty and the uncertainty due to the dim clouds while the red error bars represent the uncertainty that is caused by the conversion factors.

### 4.1.1 Accounting for differences in sensitivity

To further analyze the albedo bias between the instruments and to quantify the contribution from different sources it is useful to again consider how the dim pixels are treated in the CIPS retrieval. The CIPS sensitivity to faint clouds has been quantified by Lumpe et al. (2013) and is shown in their Fig. 18. The sensitivity to faint clouds is strongly dependent on solar zenith angle (SZA) and the detection rate is highest for high SZA and declines for lower SZA, especially for the faint clouds. The SZA during the date and time for the tomographic scans in 2010 and 2011 is around 60°. For this SZA, the detection rate for 2e-6

sr$^{-1}$ clouds is 40%, for 3e-6 sr$^{-1}$ clouds 70% and for 5e-6 sr$^{-1}$ clouds 90% (Lumpe et al., 2013). Since it is likely that CIPS reports zero cloud in the pixels where it is not able to detect clouds, the fraction of pixels in the CV element that is filled with non-zero albedo (fill factor) will inevitably affect the albedo comparison between the instruments for the faint clouds. Figure 5 compares the relative disagreement in cloud albedo between the instruments. Each dot represents the relative difference 2×(OSIRIS - CIPS)/(OSIRIS + CIPS) vs. CIPS albedo and is color-coded by the percentage of non-zero pixels in the CIPS

CV element. As can be seen from this figure, the instruments tend to disagree when the percentage of non-zero pixels decreases. For the CV elements that have a large number of dim CIPS pixels, the mean albedo for this CV element will be underestimated. The instruments agree best for bright clouds, but for faint clouds, especially where CIPS reports a mean albedo < 5e-6 sr$^{-1}$, the agreement is worse. It is also clear from Fig. 5 that for dim CIPS clouds that have a large disagreement with OSIRIS, the fill factor is small (purple color). Benze et al. (2018) evaluated the dependence of the percentage filling of the CIPS CV and

showed that the sensitivity issue could be avoided using a threshold on the fill factor of 95%. It is thus appropriate to apply a threshold on the CIPS CV fill factor in the current study, and for consistency with Benze et al. (2018) we use the value of 95%. To summarize, the following two sensitivity adjustments are applied to CIPS pixels: (1) All pixels with an albedo < 2e-6 sr$^{-1}$ are set to zero. (2) A filling factor of 95% is required, meaning that only CIPS CVs with at least 95% cloud pixels are included in the comparison. This also includes those faint pixels that were set to zero in the previous step. The few remaining non-cloud

pixels are included in the horizontal average.

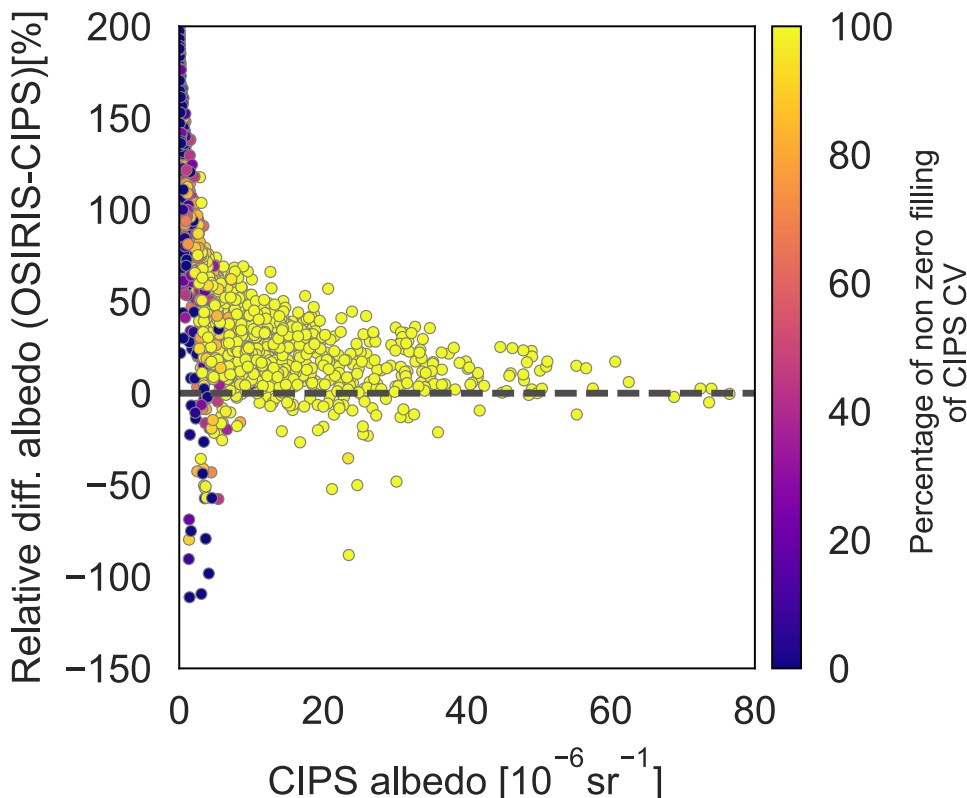

**Figure 5:** Relative difference in cloud albedo in CV elements (2×(OSIRIS-CIPS)/(CIPS + OSIRIS)) vs CIPS albedo. The points are color-coded by the percentage of filling of non-zero pixels in a CIPS CV element.

To account for the differences in sensitivity, we additionally apply a retrieval threshold on the OSIRIS cloud scattering coefficient of $10^{-10}$ m$^{-1}$sr$^{-1}$ (as discussed in Sect. 2) when vertically integrating the cloud scattering coefficient to retrieve mean albedo in the CV. All vertical levels that have a cloud scattering coefficient below the retrieval threshold will not be included in the vertical integral. Figure 6 shows a scatterplot comparing OSIRIS albedo and CIPS albedo in the common volume if we apply a fill factor threshold of 95% to CIPS CV elements and apply a retrieval threshold of $10^{-10}$ m$^{-1}$sr$^{-1}$ to OSIRIS scattering coefficient (Figure layout is the same as Fig. 4). The total number of observations decreases from 1292 to 788 due to the restriction of the filling of the CIPS CV element. Accounting for the differences in sensitivity and adding a threshold on CIPS fill factor of 95% improves the agreement between the instruments for the faint clouds. However, OSIRIS is more sensitive to smaller ice particles than CIPS, and it is possible that some of the ice that is detected by OSIRIS as faint clouds comes from clouds with small ice particles that are below the CIPS detection threshold. Some difference in the observed albedo quantities is therefore expected. With the discussed changes the observed discrepancy between the instruments increases from 2.8e-6 sr$^{-1}$ (±2.4e-6 sr$^{-1}$) (Fig 4) to 3.3e-6 sr$^{-1}$ (±2.8e-6 sr$^{-1}$) (Fig 6) and the correlation coefficient decreases from 0.97 to 0.96.

### 4.1.2 Quantitative comparison of OSIRIS IWC with CIPS

Figure 7 shows a scatterplot comparing OSIRIS and CIPS IWC in the common volume for a total set of 788 common volume element observations. Each dot represents the IWC inferred from OSIRIS and the corresponding IWC from CIPS in the CV element. The systematic uncertainty of CIPS IWC strongly increases with decreasing particle size. As described in Section 2.2.1, we take this into account by screening out the suspicious IWC detections, only including CIPS pixels that report a particle radius > 20 nm when we calculate the mean IWC in each CV element. For consistency with the albedo comparison, we have applied a CV fill threshold of 95%. For OSIRIS, the retrieval threshold ($10^{-10}$ m$^{-1}$sr$^{-1}$) is applied in the vertical integration to form IWC in the common volume by vertically integrating the IMD at each level. We find an offset between OSIRIS and CIPS IWC, and we quantify this offset (OSIRIS - CIPS) to -22 g km$^{-2}$ ±14 g km$^{-2}$.

The error bars in Figure 7 represent a total error that is relevant for the comparison between the instruments by combining systematic and statistical error from each dataset. Based on the discussion in section 2.1.1, the OSIRIS errorbars combine the absolute uncertainty of IMD due to the 10 % measurement accuracy and a statistical uncertainty that is obtained by propagating the uncertainty of the scattering coefficient through the derivation of the IMD at each level. The resulting error in IWC in the common volume element is calculated in the column integration as the combined error of IMD from all vertical levels. Similarly, the CIPS errorbars (in grey) represent the systematic and statistical uncertainty propagated from all individual CIPS pixels in the CV element. The individual uncertainty for each pixel is taken from Fig. 21 in Lumpe et al. (2013) where both the estimated systematic uncertainty and the statistical uncertainty of CIPS IWC are given as a function of IWC and particle radius for different ranges of solar zenith angles. For the range of solar zenith angles in our study (59° to 71°), the systematic uncertainty for medium (40 – 60 nm) and large (60 – 80 nm) particles is between -5 and -10 g km$^{-2}$, while for smaller particles (20 - 40 nm) the systematic uncertainty is between -5 and -35 g km$^{-2}$. In addition to the systematic uncertainty, the statistical uncertainty (standard deviation) for medium (40 – 60 nm) and large particles ( 60 – 80 nm) is between 5 and 10 g km$^{-2}$ and for smaller particles (20 – 40 nm) the statistical uncertainty is between 15 to 40 g km$^{-2}$. Especially for small particles, the statistical uncertainty is highly dependent on IWC.

Unlike for the albedo comparison (Figs. 5 and 6), where correction factors for phase and wavelength are applied, no corrections are needed for the comparison of IWC in the CV element, and hence no additional error bars for this conversion are needed. Generally, CIPS and OSIRIS IWC observations agree well within the common volume. The relative difference is large for the dimmest clouds and decreases for stronger clouds. The correlation coefficient is calculated to be 0.91 between the instruments.

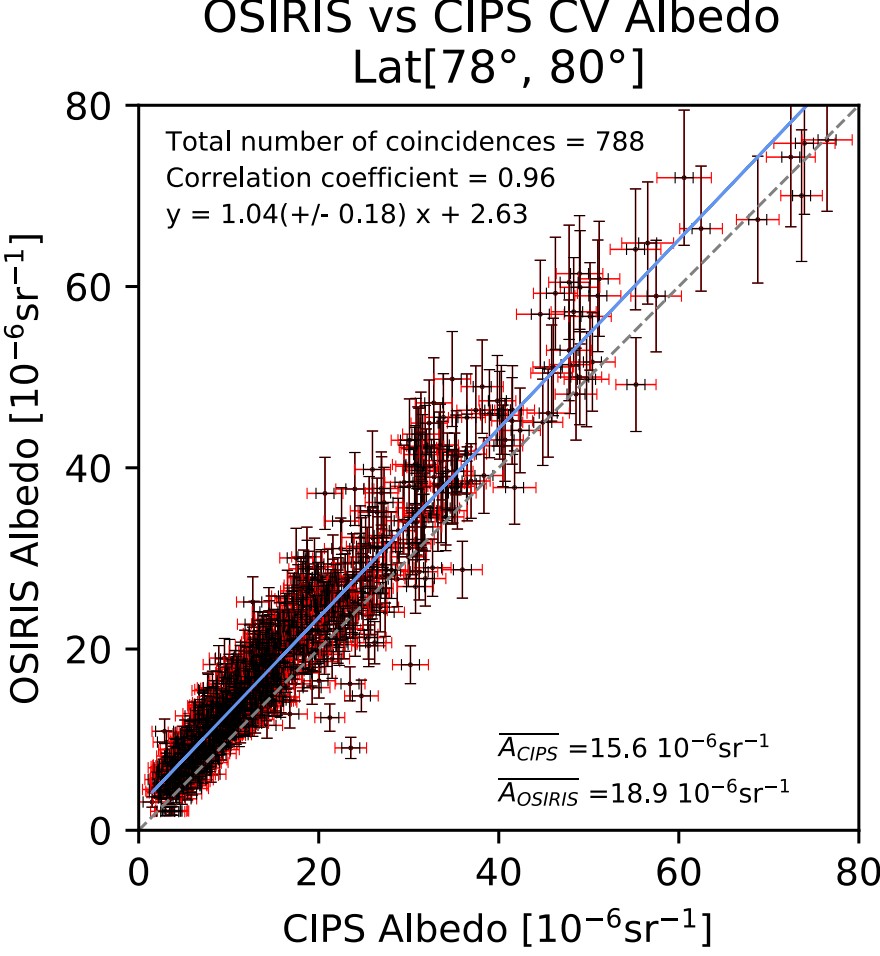

**Figure 6:** Scatterplot of OSIRIS and CIPS mean albedo in the CV. Same as Fig 4, but using thresholds on CIPS fill factor of 95% and OSIRIS scatter coefficient threshold of 10$^{-10}$ m$^{-1}$sr$^{-1}$ as discussed in the text. The average albedo for CIPS and OSIRIS for all the CV in the figure is indicated in the bottom right of the figure. The grey dashed line denotes the one-to-one line. The blue line is the regression line.

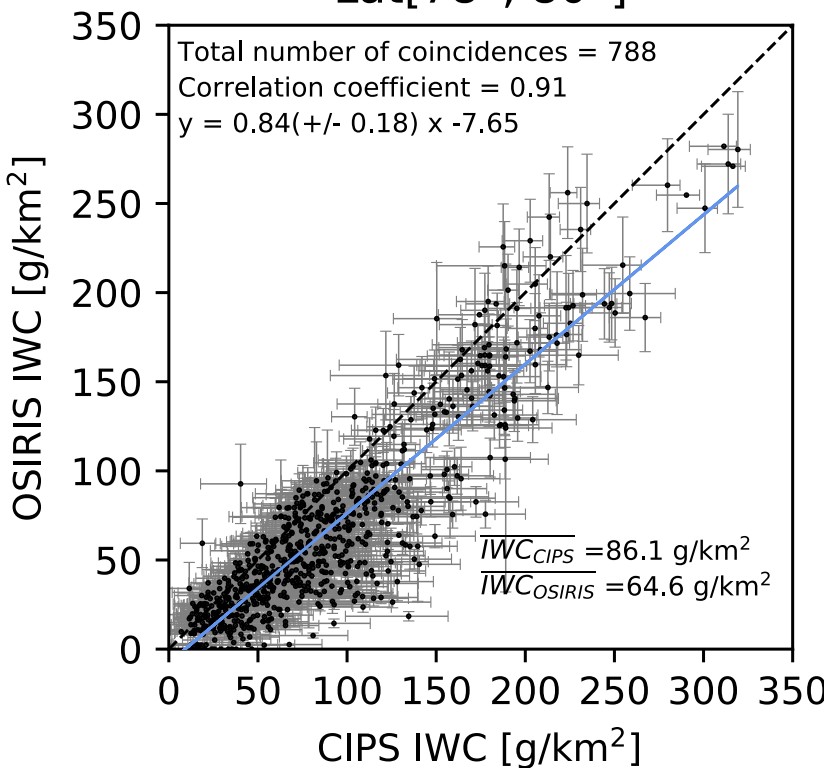

**Figure 7:** Scatterplot of OSIRIS and CIPS common volume ice water content. The average IWC for CIPS and OSIRIS is indicated in the bottom right of the figure. The error bars are a combination of the systematic and statistical uncertainty from each instrument, thus representing a total uncertainty that is relevant when comparing the datasets. The grey dashed line denotes the one-to-one line. The blue line is the regression line.

## 4.2 Orbit-wise comparison of OSIRIS albedo with CIPS

The results from the above statistical comparison show that the instruments agree very well on average, that the choice of method used for the instrument comparison is valid, and that the time constraint and size of the common volume are suitable. In this section, we continue to compare the instruments using the individual strengths of each instrument, i.e. the horizontally resolved CIPS data and the vertically resolved OSIRIS data. As described in Section 2.1, the ability of OSIRIS to resolve individual cloud structures varies stochastically. As the number of measurements along the orbit is sparse, the resolution depends on the placement of lines of sight relative to the actual cloud structure. Therefore, when comparing CIPS data to the OSIRIS tomography in this section, we apply a horizontal integration of the CIPS data of both 56 km (size of OSIRIS retrieval pixel) and 280 km (typical OSIRIS averaging kernel) as limiting cases.

We present the observations of cloud brightness from three individual OSIRIS orbits and the coinciding CIPS orbits in Figs. 8, 9, and 10. These particular orbits were chosen to illustrate some examples of when the cloud in the CV show good agreement, and point out one example when the cloud observations in the CV disagree, and thus illustrate for the reader the range of cloud observations available for this study. The top panel of Figs. 8, 9 and 10 show the vertically and horizontally resolved OSIRIS scattering coefficient for the subset of the orbit that overlaps the coinciding CIPS orbits strip. The abscissa is given in OSIRIS Angle Along Orbit (AAO) and the vertical axis covers the subrange 78-88 km. The second panel shows CIPS albedo for the subset of pixels that overlap the OSIRIS field of view along the OSIRIS line of sight. The horizontal pink lines mark the width of OSIRIS field of view, and the CIPS pixels within these lines denote the pixels within the CV. Note that this plot is not the normal CIPS orbit strip image that uses polar projection map, but only the subset of pixels in the CIPS CV level 2 geolocated

data that we have plotted on a new grid to facilitate a comparison using the same abscissa. The region in the close proximity (+/-30 km, i.e +/-6 CIPS pixels) of the CIPS common volume is included in this figure for illustrative purposes. The abscissa in the middle panel is given in CIPS pixels along OSIRIS LOS, hence identical with the abscissa in the top panel. The bottom panel shows the resulting albedo in the CV for each instrument given in the unit G and along the abscissa AAO. The black

line shows the mean OSIRIS albedo in the common volume corresponding to the scattering coefficient in the first panel. The purple line shows the mean CIPS albedo in the common volume when a horizontal integration of 56 km is applied (high-resolution limit) and the green line shows the mean CIPS albedo in the common volume when a horizontal integration of 280 km is applied (low-resolution limit).

For OSIRIS orbit 51236 (Fig. 8) observed during July 16, 2010, a bright continuous cloud layer is visible between approximately 82.5 and 85 AAO with the brightest region of the cloud at 84-84.5 AAO and at altitude ~83.5 km. Simultaneously, the coinciding CIPS orbit 17556 observes the same bright cloud layer, although the higher horizontal resolution of CIPS makes it possible to detect the bright cloud region at 83.5 AAO. The higher horizontal resolution of CIPS makes it possible for this instrument to clearly distinguish this bright cloud region even averaged over the horizontal extent of

one CV, which can be seen by the peak of ~50e-6 sr$^{-1}$ at 83.5 AAO in the third panel. Applying the averaging kernel to CIPS (and thus "smooth" the data to OSIRIS resolution) produces a mean albedo of ~32e-6 sr$^{-1}$ in the same cloud region, very close to the OSIRIS albedo of ~28e-6 sr$^{-1}$. Comparing the mean albedo from CIPS and OSIRIS in this orbit confirms the good agreement concerning both the absolute value of the peak albedo and the spatial variations of the albedo throughout the extent of the overlapping cloud volumes.

For OSIRIS orbit 50796 (Fig. 9) observed on June 17, 2010, a cloud layer is visible between 81-83 AAO with the brightest cloud layer at an altitude of 85 km at 81.5 AAO. Coinciding CIPS orbit 17117 shows a cloud in the same region that is highly structured in the CV. By comparing the mean albedo in the CV, we note that OSIRIS albedo is 2-5e-6 sr$^{-1}$ higher than CIPS (before applying the averaging kernel) throughout the whole CV. The observed differences for this orbit are larger than the estimated error bars.

For OSIRIS orbit 51646 (Fig. 10) taken during July 16, 2010, a bright cloud layer with a vertical extent of ~4 km encompass the whole CV from AAO 81 to 85. The brightest cloud region is found at an altitude of 84.5 km at AAO 82.5. Simultaneously, the coinciding CIPS orbit 17965 observes the same bright cloud layer, but additionally reveals that this cloud layer is highly structured. The brightest cloud layer observed by CIPS is located at 83-83.5 AAO. In this orbit, the resulting mean albedos do not show good agreement, and an apparent horizontal shift between OSIRIS and CIPS is noteworthy. In cases like this, with a

sharp gradient across the OSIRIS LOS, a possible explanation for the discrepancy is the action of horizontal wind transporting clouds out of or into the CV. At PMC altitude the horizontal wind is mainly modulated by atmospheric tidal waves and gravity waves. Horisontal wind velocities of ~50 m/s have been observed by The Midle Atmosphere Alomar Radar System (MAARSY) in the northern Norway (Stober et al., 2013). In this CV, the time difference between CIPS-OSIRIS observations is ~5 minutes During this time, a wind speed of 50 m/s would transport a cloud 15 km, which corresponds to the half width of

OSIRIS LOS. It is likely that the cloud region outside the CIPS-OSIRIS CV (as indicated by the pink lines in the third panel) could have been transported into the CV by wind directed across the OSIRIS LOS.

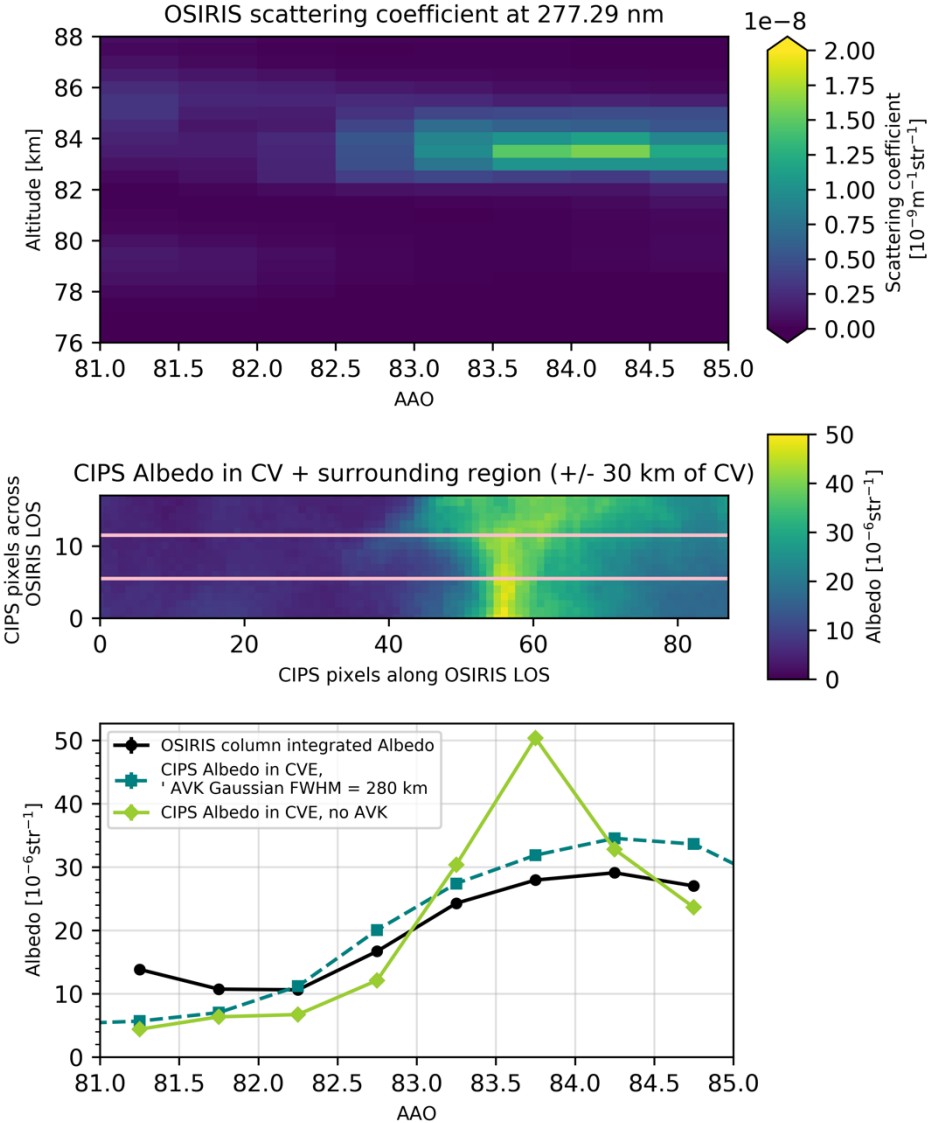

**Figure 8:** Common volume observations of cloud albedo for OSIRIS orbit 51236 and CIPS orbit 17556. The coincidence occur at latitude 78°N and longitude 150° at 15.45 local time. The top panel shows OSIRIS cloud scattering coefficient (vertical plane) for the subset of the orbit that overlaps CIPS orbit strip. The middle panel shows CIPS albedo (horizontal plane) for the subset of pixels that overlap OSIRIS field of view along OSIRIS LOS. The pink lines denote the width of OSIRIS LOS, and consequently the CIPS pixels within the pink lines that are contained in the CV. The region +/-30 km outside the CV is included only for illustrative purposes. The abscissa in the middle panel is given in CIPS pixels along OSIRIS LOS, hence identical with the abscissa in the top panel. The bottom panels shows the resulting albedo in the CV for the integrated signal from each instrument along the abscissa AAO.

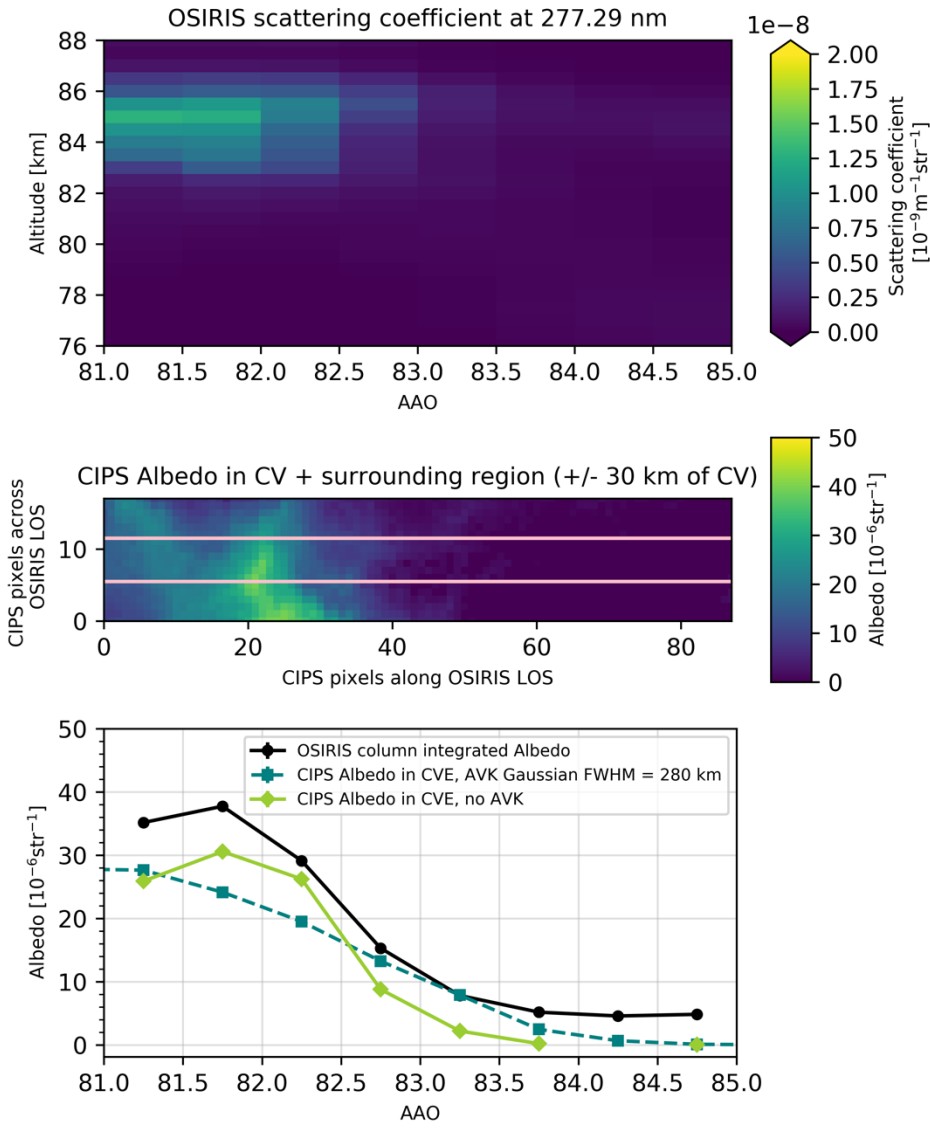

**Figure 9**: As Fig. 8, but for OSIRIS orbit 50796 and CIPS orbit 17117. The coincidence occur at latitude 79°N and longitude 305° at 15.45 local time.

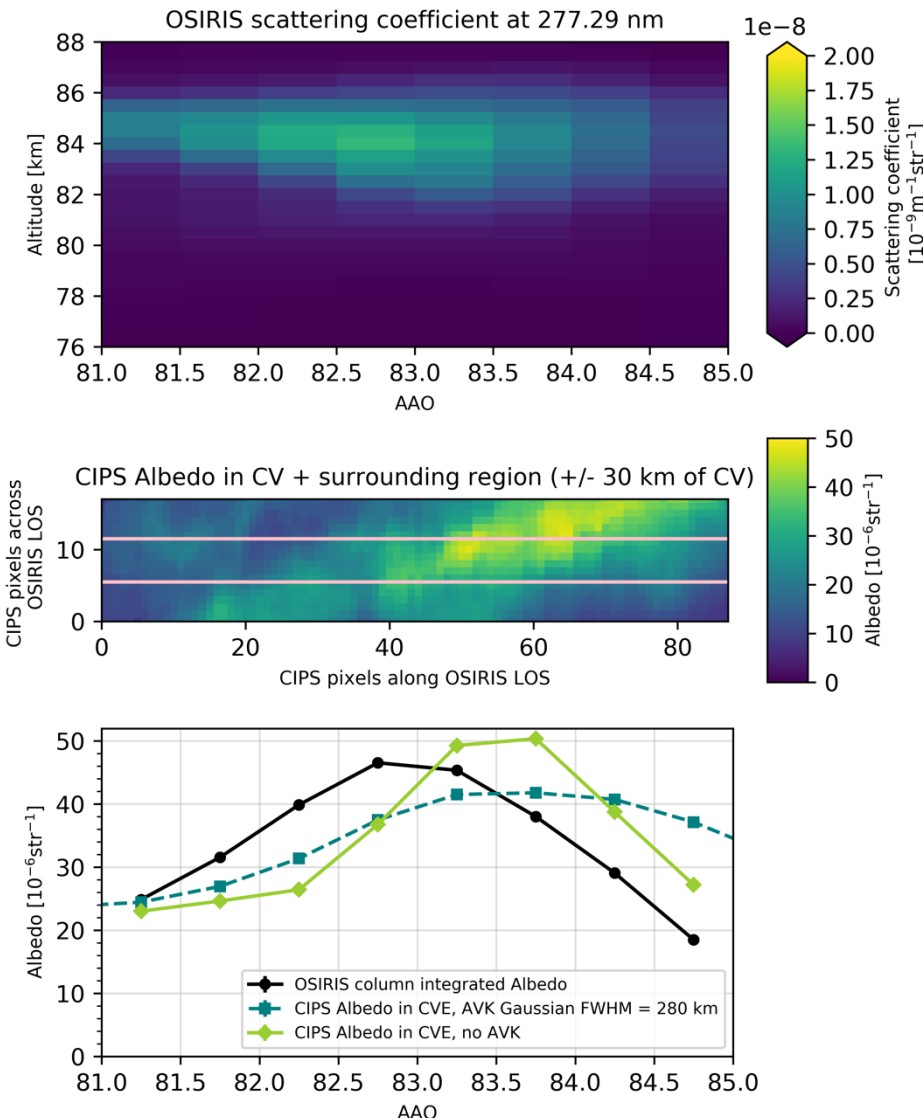

**Figure 10**: As Fig. 8, but for OSIRIS orbit 51646 and CIPS orbit 17965. The coincidence occur at latitude 79°N and longitude 357° at 15.45 local time.

**5. Discussion and Conclusions:**

In this study, we have compared the PMC cloud properties cloud albedo and IWC from Odin OSIRIS limb tomography to the nadir viewing AIM CIPS. The analysis is performed for northern hemisphere 2010 and 2011 for a total set or 180 coinciding orbits at latitudes from 78N to 80N for local times ~ 15.45. The OSIRIS tomographic PMC dataset provides combined coarse horizontal and high vertical information, while CIPS provides preeminent horizontal PMC information. When combined in a common volume study, OSIRIS can provide vertical information of structures such as double cloud

layers and ice voids as well as detailed particle size and number concentration information at variuos height levels to the detailed horizontal PMC information from CIPS. This information can be used to study how atmospheric waves of different scales (inferred from albedo variations in CIPS) alters the vertical distribution of cloud properties (inferred from OSIRIS).

Additionally to such studies, the combined CIPS/tomographic dataset OSIRIS provide useful insight to more detailed studies of the PMC particle size distribution.

First, we have extended the previous OSIRIS error description by Hultgren et al. (2013) by performing a detailed error characterization for local cloud scattering coefficient and IMD that takes into account absorption of mesospheric ozone along the LOS. Second, we have compared these cloud properties to common volume observations from CIPS. To be able to compare the common volume cloud properties from two different satellite instruments, OSIRIS using limb geometry and adapting spectroscopy to retrieve cloud properties, and CIPS using nadir geometry and adapting multi-angle phase function observations of PMC to retrieve cloud properties, it is necessary to account for the differences in scattering conditions, observational volume, and sensitivity. In this study, we have averaged the high horizontal resolution CIPS albedo and IWC to the coarser horizontal resolution of OSIRIS tomography. Additionally, we have vertically integrated OSIRIS scatter coefficient and IMD to obtain albedo and IWC comparable to CIPS. We have accounted for the differences in scattering conditions by transforming CIPS albedo into the SSA and wavelengths used by OSIRIS. By adopting a very narrow spatial and temporal coincidence criterion, we have been able to capture a large variety of albedo and IWC in the common volume. We have shown that OSIRIS error characterization of volume scatterring coefficient and IMD is valid by demonstrating that the cloud properties within the common volume largely agree within the specified error for each instrument analysis. We find that OSIRIS cloud scattering coefficient shows excellent agreement with CIPS cloud albedo with a correlation coefficient of 0.96, although in the common volume OSIRIS observes brighter clouds than CIPS. The bias between the instruments is found to be 3.4e-6 $sr^{-1}$ ($\pm$ 2.9e-6 $sr^{-1}$). The correlation is higher for bright clouds than for faint clouds. Owing to differences in instrumental design and geometry, the instruments have different sensitivity. We account for these differences by adding a threshold on CIPS CV fill factor of 95% and a scatter coefficient threshold to OSIRIS of $10^{-10}$ $m^{-1}sr^{-1}$ and find that this leads to a better agreement for faint cloud volumes. Additionally, we find good agreement between OSIRIS IMD and CIPS IWC, with a correlation coefficient of 0.91. However, CIPS observes more ice than OSIRIS in the common volume, and the bias (OSIRIS-CIPS) is found to be -22 g $km^{-2}$ ($\pm$ 14 g $km^{-2}$).

A reason for why OSIRIS IWC is biased low to CIPS IWC in is considered to arise from how ice mass density (IMD) is calculated in OSIRIS PMC retrieval. When calculating IWC from the vertical integration of OSIRIS IMD data, we currently only take into account retrieval pixels that are bright enough so that a spectroscopic size retrieval is feasible. It is possible that the OSIRIS analysis misses ice from weak pixels where mean radii smaller than 20 nm are reported. Such weak cloud pixels cover typically an altitude range of 1 km in the upper part of the cloud. A rough estimate of how much ice OSIRIS misses by ignoring such pixels can be given by considering how much ice can be produced when converting a typical concentration of water vapour at 86 km (e.g., 3 ppm water vapour in 2e14 $cm^{-3}$ air) into ice. Calculating this, a 1 km thick layer contributes to the overall IWC with 18 g $km^{-2}$. This is more or less the difference that we see in Figure 7 between OSIRIS and CIPS IWC. Bailey et al. (2015) compare albedo, radius, and IWC in a CIPS-SOFIE CV that is located at high SZAs. These clouds occur right at the terminator, where the Rayleigh background is small and has a large slope vs. SZA. This is a difficult region to observe PMCs, as even a small error in the CIPS Rayleigh background is expected to have a large impact on the retrieved cloud products. Bailey et al. (2015) apply a justifiable correction to the CIPS background removal which brings the SOFIE and CIPS observations, specifically albedo and IWC, into better agreement than without (their Figure 11). The present study uses cloud observations at a much more favorable SZA range (59-71 deg). Here, CIPS background changes such as found by Bailey et al. (2015) are negligible, and no CIPS background correction is necessary. Bailey et al. (2015) show that the reported v4.2 CIPS IWC is 50% smaller than SOFIE IWC in the common volume, but only ~30% smaller when correcting the CIPS background removal (their Figure 11). In the present study, we find that CIPS IWC is 33% ($\pm$22%) greater than that of OSIRIS in the common volume (Figure 7). This difference between the two studies is important because IMD is the native measurement quantity for SOFIE, which can thus be seen as a reference measurement. SOFIE uses the technique of satellite solar occultation to measures vertical profiles of limb path atmospheric transmission,

and as such is much more sensitive to small ice particles than OSIRIS. The two studies use different methods: Bailey et al. (2015) model what CIPS should observe based on SOFIE observations, whereas this study corrects for sensitivity differences and compares the horizontally and vertically averaged quantities. Another difference is that this study uses observations at more favorable SZAs and higher latitudes, therefore avoiding uncertainties of the CIPS background and containing generally brighter clouds with more ice.

An additional reason for why OSIRIS IWC is biased low to CIPS IWC in this work is considered to arise from how ice mass density (IMD) is calculated in OSIRIS PMC retrieval. When calculating IWC from the vertical integration of OSIRIS IMD data, we currently only take into account retrieval pixels that are bright enough so that a spectroscopic size retrieval is feasible. It is possible that OSIRIS miss ice from the pixels where a mean radius > 20 nm are reported. A rough estimate of how much ice OSIRIS miss by ignoring weak cloud pixels that cover typically an altitude range of 1 km in the upper part of the cloud can be given by considering how much ice we can produce when converting a typical concentration of water vapour at 86 km (e.g., 3 ppm water vapour in 2e14 cm-3 air) into ice. Calculating this, a 1 km thick layer contributes to the overall IWC with 18 g km-2. This is more or less the difference that we see in Figure 7 between OSIRIS and CIPS IWC.

We have also performed a more detailed comparison of cloud properties in individual orbits. The sample of orbits used in this study illustrates the strength of this dataset. The vertically resolved OSIRIS tomography in combination with the horizontally high resolved CIPS data provide tools to study cloud structures, and how these, in turn, affect the observed cloud properties in the CV.

This study has validated the OSIRIS tomographic PMC cloud brightness and ice content against nadir-viewing CIPS observations. It has addressed the potential errors from the dataset itself, as well as errors inherent to the comparison of limb tomography and nadir PMC retrievals. Due to the limitation of a total of 18 days of observations during the seasons NH 2010 and NH 2011, we have not performed a detailed comparison of cloud frequency. A follow-up paper is planned to discuss how to best compare OSIRIS limb tomography and CIPS column-integrated data when it comes to PMC particle size retrievals.

**Data availlability**

The Odin OSIRIS tomography PMC dataset used in this article, as well as codes used for the analysis are available upon request to L. Broman. AIM CIPS data is available for download from the Laboratory for Atmospheric and Space Physics webpage: http://lasp.colorado.edu/aim/download-data.php

**Author contributions**

Jörg Gumbel and Susanne Benze  presented the original idea of this study. Lina Broman designed the study, carried out the anlysis and wrote the major part of the manuscript. Susanne Benze and Ole-Martin Christensen were responsible for processing the tomographic OSIRIS data. Ole-Martin Christensen and Jörg Gumbel perfomed tests of the resolultion of the OSIRIS tomography. All co-authors contributed to the interpretation of the results, provided feedback and contributed equally to the improvement of the manuscript.

**Competing Interests**

The authors declare that they have no conflict of interest.

**Acknowledgements**

We thank Jerry Lumpe for valuable discussion related to the CIPS retrievals. We thank Gerd Baumgarten for providing scattering data from T-Matrix calculations. We thank Matthew DeLand for the helpful duscussions regarding section 5. We are grateful to the two anonymous reviewers whose comments have greatly imprved this manuscript. Odin is a Swedish-led satellite funded jointly by Sweden (SNSB), Canada (CSA), France (CNES), and Finland (TEKES). Since 2007, Odin is a third-party mission of the European Space Agency. AIM was developed as part of NASA's Small Explorer series of missions. We gratefully acknowledge the efforts of the entire Odin and AIM development, science, and operations teams.

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
