# Peer review of "Common volume satellite studies of polar mesospheric clouds with Odin/OSIRIS tomography and AIM/CIPS nadir imaging"

_Atmospheric Chemistry and Physics, 2018_

## Referee Comment (RC1) · Anonymous Referee #2 · 25 Feb 2019

General Comments

This paper presents a detailed comparison of polar mesospheric cloud (PMC) albedo and ice water content between two different satellite instruments. The instruments are the Optical Spectrograph and InfraRed Imager System (OSIRIS) on the Odin satellite and the Cloud Imaging and Particle Size (CIPS) instrument on the Aeronomy of Ice in the Mesosphere (AIM) satellite. Because OSIRIS typically views PMCs on the limb whereas CIPS typically views PMCs in the nadir, the authors have carefully considered coincidence criteria, scattering conditions, observation geometry, and instrument sensitivity in the uniquely coordinated study between the two instruments. As part

of the study, the authors present the first thorough error characterization of OSIRIS tomographic cloud brightness and ice water content.

This is an important paper and establishes a valuable precedent for subsequent comparisons between PMC limb viewing instruments and PMC nadir imagers. The results show good agreement, particularly given the diversity of data included in the study. Most importantly, however, the authors provide an exhaustive error analysis that will be a useful reference in future PMC correlative studies.

The Reviewer recommends the paper for publication provided that the authors address the comments below. The "Specific Comments" are relatively minor but important, particularly in providing context of their results with the existing body of work on this topic.

Specific Comments

1. Abstract. Please indicate latitude range and years used in the analysis. Also, if PMC frequency is not compared between OSIRIS and CIPS within the common volume, the authors should explicitly say so in the abstract.

2. p. 8. Lines 9-11. Did Benze et al. [2011] use the operational CIPS product to compare directly with SBUV? The Reviewer looked at this paper and it appears that the good agreement with SBUV as stated here arises because a separate CIPS retrieval was developed to simulate the SBUV PMC retrieval. This is not a validation of the CIPS or SBUV data, which is what is suggested by this statement. How do operational SBUV and v4.20 CIPS PMC albedos, IWC and frequencies compare for the same volume and the same time at these high latitudes (78-80 N)? If the authors do not have a ready answer or if it is beyond the scope of this work then they should be explicit about what was done previously to find agreement between CIPS and SBUV (i.e. a separate CIPS algorithm). They could also delete these sentences entirely without loss of content to the paper.

[Figure]

3. p. 16, end of section. Please include a paragraph here explicitly indicating what is done with pixels where there are no clouds at all. If the authors have set all pixels less than 2e-6 sr-1 to zero, then they need to explicitly say here whether they have averaged the zeros into their calculations of albedo and IWC or not. This distinction has historically been a source of great confusion in the field of PMCs. It may be that at these high latitudes there is always a cloud within the common volume for their limited dataset and if so they need to say that as well. This does not appear to be the case from looking at Figure 2. However, the authors winnow the dataset to 788 total observations (p. 17, line 9) so it is not clear how Figure 2 evolves with the study.

4. Conclusions. This section is lacking a summary of relevant conditions under which the comparisons are made. This includes (but is not limited to) the years studied, the latitudes used and the local times of the comparisons. This should also emphasize that the authors are comparing albedo and IWC and not PMC frequency. This section is also lacking a summary of previous related work by Bailey et al. [2015] using common volume observations of SOFIE and CIPS on the same AIM satellite. Bailey et al. state that CIPS IWC is a factor of two smaller than SOFIE IWC, differences that are generally larger and go in the opposite direction of the present work with OSIRIS. Although the authors have done a thorough analysis of their two datasets (OSIRIS and CIPS), the reader should be made aware of these differences of CIPS IWC with the limb viewing SOFIE IWC. This is all the more important because IWC is the native measurement quantity for SOFIE. Differences in the method of observation, calibration, coincidence criteria, latitudes of the comparison, the years studied, the solar zenith angles and the local times of the comparison may all play a role in reconciling these differences and could be included to raise awareness with the reader. The above could be done with two paragraphs and if the authors prefer they could rename this section "Discussion and Conclusions".

Technical Corrections

p. 1, line 17. "ice" should be "ice water content". Similarly, on line 20 "ice content"

should be "ice water content" to avoid confusion.

p. 2, line 17. "larg" should be "large"

p. 2, line 30. "…reason for the increasing visibility of PMCs at mid-latitudes" should be "…reason for the increasing visibility of PMCs at mid-latitudes in the modern era." To the Reviewer's knowledge, decadal-scale trends of mesospheric clouds observed from the ground since the late 20th century are weak or non-existent.

p. 2, line 36. "advantage" should be "advantages".

p. 10, Figure 1. This figure has a geographic range that is much larger than the region of interest and could be improved dramatically. On lines 9-10 the authors say that the cloud albedo is variable but the region of interest is drawn over the data so the reader cannot see this. By reducing the latitude and longitude ranges of the image, only the borders of the region of interest can be drawn and the boxed region can remain unfilled so that the reader can see the structure within. If the geographic range is small enough, the red area could also be drawn with borders rather than filled. If the authors prefer, the figure could be drawn with two panels: Panel "a" could be the current figure and panel "b" could be the zoomed in version. Please also include a color bar showing the range of cloud albedo in the figure(s).

p. 11, Figure 2. Please include tick marks on the x and y axes to better guide the reader. Also, please indicate the total number of detections either within the Figure or in the caption. Since this is the first figure quantitative showing CIPS data, it would also be instructive to indicate that this is CIPS data, and include average latitude, local time, year of the data and a CV frequency either within the Figure or in the caption. Thank you.

p. 13, line 15. Please include here the ranges of Cspectral and Cphase used in the analysis so that the reader can appreciate the impact of these adjustments in the context of the data.

p. 14. The offset and uncertainty in line 17 is a bit different than line 26. This is further modified on p. 17 line 15, but is still not quite the same as reported in the abstract and summary. Please check to make sure the numbers self-consistent throughout. Thank you.

p. 15, Figure 4 caption. "error bars is" should be "error bars are" and "error bar denote" should be "error bars denote".

p. 16 lines 2-4. Do the authors mean the error bars in Figures 4 and 6? Please indicate the figures explicitly. Also, the Reviewer only sees black (not grey) error bars in these figures. Are they referring to these? Please be explicit. Thank you.

p. 20, line 30. A wind of 100 m/s near 85 km seems large. Can the authors provide a reference for this or otherwise justify this wind speed? Is it possible that the cloud could be sublimating and reforming elsewhere? If so the authors should state that as a possibility. To this end, the authors should include the time difference between the two observations in the captions of Figures 8, 9 and 10.

p. 24, lines 25-27. Please explicitly indicate the version of CIPS data used in this study here (in addition to p. 8, line 8).

Figures 4, 6 and 7. Please indicate explicitly whether null detections are included in the indicated average. If null detections are ignored in these comparisons then they should say that instead.
* * *

---

## Referee Comment (RC2) · Anonymous Referee #1 · 1 Apr 2019

Limb and nadir viewing satellite observations have become key observational methods for studying the physical processes leading to the formation and driving the variability of polar mesospheric clouds (PMCs). This study is the first to undertake a systematic comparison of limb tomography and nadir observations of PMCs in real common volumes. Both directly observable quantities such as cloud albedo and scattering coefficients are compared as well as inferred cloud properties such as ice mass density and ice water content. Importantly, this is done by thoroughly taking into account the effects of scattering geometry, differences in spatial resolution, as well as - and most importantly - the detailed error budget of the different observations.

Given the uniqueness of the two considered data sets as well as the great scientific interest in PMC processes this is an important study that paves the way for future applications of this combined data set for studies into the fundamental properties of PMC.

As such I am recommending acceptance of this manuscript provided that the following mostly minor comments are adequately addressed:

- Abstract: the statement that ice mass densiy agrees with ice water content doesn't make sense since these are two totally different quantities (one is the other integrated in the vertical). Of course, when reading the full text it is clear that the authors mean that the two properties are consistent with each other after properly accounting for the vertical extent of the cloud and integrating the limb observations in the vertical. Please clarify.

- Page 1, line 35: Gadsden and Schröder is a nice textbook but certainly not an original scientific reference. Please replace with suitable references of original measurements (e.g., Lübken, 1999 and/or some even older papers from the Stockholm group based on rocket grenade measurements).

- Page 2, line 4: When referring to these initial observations reference should also be made to the paper by Jesse, 1885:

Jesse, O., Auffallende Erscheinungen am Abendhimmel, Met. Zeit., 2, 311-312, 1885.

- Page 2, line 8: Please add "e.g.," in front of the reference to the paper by Fritts et al., 1993.

- Page 2, line 9: At the end of the sentence after the reference above, I would add a reference to the classical paper by Witt, 1962: Witt, G., Height, structure and displacements of noctilucent clouds, Tellus XIV , 1 , 1-18, 1962.

- Page 2, line 22: typo "ferquency"

- Page 2, line 26: How can the ALOMAR lidar data allow to make statements on the horizontal extent of clouds?

- Page 2, line 39: Delete "systems"?

- Page 3, line 41: Maybe clarify that you are discussing the operational retrievals here.

- General statement to introduction: I am missing a paragraph pointing out what added value the combination of nadir and limb data sets allows to benefit from. The three aims listed on page 4 are all quite technical; in order to make this manuscript fit to the scope of a scientific journal such as ACP, the readers should stress the added value of the combined data set and explain what kind of studies can be better done with the combined data set than with the single data sets alone. A similar statement should also be added to the conclusions/summary and the abstract.

- Page 6, line 20: This these -> These

- Page 7, line 28: Please consider adding a table that summarizes the uncertainties that have been mentioned in the text above. Ideally the corresponding uncertainties from CIPS should also be included to make this discussion easier to follow.

- Page 9, line 28: Do you really mean systematic error or just systematic deviation/difference?

- Figure 2: I am missing a discussion of the shape of this distribution. Is it coincidence that the distribution is roughly symmetric around 50%?

- Figure 3: Excellent!

- Page 13, line 7: Reference to Baumgarten et al., ACP 2010 should be added.

- Page 13, line 17: Please spell out "IMD".

- Figures 4,6, and 7: haven't you regressed these data sets? I recommend to show the regression lines and indicate the corresponding parameters with their error bars.

[Figure]

- Figure 4: Why is the OSIRIS error bar increasing with OSIRIS albedo?

- Page 19, line 15/16: clearly three examples from such a large data set cannot be representative. At most they may illustrate the class of comparisons. Please reword.

- Page 19, line 22: that are we have -> that we have

---

## Author Response (AR1)

**Reply to comments by reviewer #1**

**The comments from the reviewer are written in standard font. The comments from the authors are written in bold font. All page numbers and line numbers refer to the original draft.**

Limb and nadir viewing satellite observations have become key observational methods for studying the physical processes leading to the formation and driving the variability of polar mesospheric clouds (PMCs). This study is the first to undertake a systematic comparison of limb tomography and nadir observations of PMCs in real common volumes. Both directly observable quantities such as cloud albedo and scattering coefficients are compared as well as inferred cloud properties such as ice mass density and ice water content. Importantly, this is done by thoroughly taking into account the effects of scattering geometry, differences in spatial resolution, as well as - and most importantly - the detailed error budget of the different observations. Given the uniqueness of the two considered data sets as well as the great scientific interest in PMC processes this is an important study that paves the way for future applications of this combined data set for studies into the fundamental properties of PMC. As such I am recommending acceptance of this manuscript provided that the following mostly minor comments are adequately addressed:

**Reply: We thank the reviewer for this encouraging comment and positive feedback to our study. The detailed comments by the reviewer have definitely led to an improvement of the manuscript, and we appreciate the effort put into these comments.**

Abstract:
The statement that ice mass density agrees with ice water content doesn't make sense since these are two totally different quantities (one is the other integrated in the vertical). Of course, when reading the full text it is clear that the authors mean that the two properties are consistent with each other after properly accounting for the vertical extent of the cloud and integrating the limb observations in the vertical. Please clarify

**Reply: Good point. We replaced the text section:**

**"We find that the primary OSIRIS tomography product, cloud scattering coefficient, shows very good agreement with the primary CIPS product, cloud albedo with a correlation coefficient of 0.96. However, OSIRIS systematically reports brighter clouds than CIPS and the bias between the instruments (OSIRIS - CIPS) is 3.4e-6 sr$^{-1}$ (±2.9e-6 sr$^{-1}$) on average. The OSIRIS tomography ice mass density agrees well with the CIPS ice water content, with a correlation coefficient of 0.91. "**

**With the following text section:**

**"We find that the OSIRIS albedo (obtained from the vertical integration of the cloud scattering coefficient) shows very good agreement with the primary CIPS product, cloud albedo with a correlation coefficient of 0.96. However, OSIRIS systematically reports brighter clouds than CIPS and the bias between the instruments (OSIRIS - CIPS) is 3.4e-6 sr$^{-1}$ (±2.9e-6 sr$^{-1}$) on average. The OSIRIS tomography ice water content (obtained from the vertical integration of ice mass density) agrees well with the CIPS ice water content, with a correlation coefficient of 0.91. "**

- Page 1, line 35: Gadsden and Schröder is a nice textbook but certainly not an original scientific reference. Please replace with suitable references of original measurements (e.g., Lübken, 1999 and/or some even older papers from the Stockholm group based on rocket grenade measurements).

**Reply: This has been corrected, the reference Gadsden and Schröder has been replaced by Lübken, 1999.**

- Page 2, line 4: When referring to these initial observations reference should also be made to the paper by Jesse, 1885: Jesse, O., Auffallende Erscheinungen am Abendhimmel, Met. Zeit., 2, 311-312, 1885.

**Reply: The reference (Jesse, 1885) has been added.**

- Page 2, line 8: Please add "e.g.," in front of the reference to the paper by Fritts et al., 1993.

**Reply: This has been added.**

- Page 2, line 9: At the end of the sentence after the reference above, I would add a reference to the classical paper by Witt, 1962: Witt, G., Height, structure and displacements of noctilucent clouds, Tellus XIV , 1 , 1-18, 1962.

**Reply: Reference added.**

- Page 2, line 22: typo "ferquency"

**Reply: Corrected.**

- Page 2, line 26: How can the ALOMAR lidar data allow to make statements on the horizontal extent of clouds?

**Reply: This reference had been misquoted by the authors and we thank the**

reviewer for pointing this out. The following whole sentence was deleted: "Moreover, their study also contained detailed observations of changes in the horizontal extent of PMCs at different altitudes; specifically, they were able to demonstrate that the altitude of faint clouds decreases during the 22-year period."

- Page 2, line 39: Delete "systems"?

Reply: Corrected.

- Page 3, line 41: Maybe clarify that you are discussing the operational retrievals here.

Reply: This clarification has been added, and the sentence has been changed to : "Another advantage is that the same assumption regarding the mathematical shape of the particle size distribution, namely a Gaussian distribution, is used in both the OSIRIS and the operational CIPS v4.2 retrieval"

- General statement to introduction: I am missing a paragraph pointing out what added value the combination of nadir and limb data sets allows to benefit from. The three aims listed on page 4 are all quite technical; in order to make this manuscript fit to the scope of a scientific journal such as ACP, the readers should stress the added value of the combined data set and explain what kind of studies can be better done with the combined data set than with the single data sets alone. A similar statement should also be added to the conclusions/summary and the abstract.

Reply: We agree, thanks for pointing this out. To address this issue, we updated the text section on p. 3, line 32 - p. 4 line 16 to the following:

A comparison of the two instruments is therefore ideally suited for instrument validation and the combination of the two datasets will be valuable in future studies of cloud-wave interaction, studies on particle sizes as well as studies on how the retrieved clouds properties are affected by cloud inhomogeneit**ies**. Many scientific questions about the PMC lifecycle are connected to the 2- or 3-dimensional structure of the clouds. Important such questions concern e.g. the effect of gravity waves or dynamical instabilities on the growth, sublimation or appearance of the clouds. Combined observations by (horizontally resolved) nadir instruments and (vertically resolved) limb instruments have a large potential of addressing such multi-dimensional questions. This is true in particular if the datasets involve tomographic analysis, as in the case of the OSIRIS data utilized here.

Taking into account that the satellites have different viewing geometry, resolution and sensitivity, we analyze cloud brightness and the cloud ice in the CV and perform a detailed error analysis. One advantage of comparing

tomographic OSIRIS observations to CIPS observations is that both instruments measure scattered radiance, although OSIRIS measures with limb-viewing geometry and CIPS uses nadir-viewing geometry. Another advantage is that the same assumption regarding the mathematical shape of the particle size distribution, namely a Gaussian distribution, is used in both the OSIRIS and the operational CIPS v4.2 retrieval.

The specific aims of this satellite comparison study are:

1. Perform the first thorough error characterization of the Odin OSIRIS tomographic dataset.

2. Validate the tomographic retrieval and error characterization by comparing PMC albedo and ice water content from the Odin/OSIRIS retrievals and AIM/CIPS PMC retrievals.

3. Establish a consistent method for comparing cloud properties from a limb sounding tomographic data set to a nadir viewing instrument.

4. Produce a combined dataset of Albedo and Ice water content that will facilitate future studies of the PMC lifecycle and PMC particle sizes.

This study focuses on comparing albedo and ice water content between the instruments. A future goal is to produce a combined dataset that can be used to study for example more fundamental issues such as the assumption of the PMC size distribution, an assumption that has been questioned in the past. Each instrument used alone can only provide either fine horizontal resolution (CIPS) or vertical/coarse horizontal resolution (tomographic OSIRIS). However, when combined in an efficient way, OSIRIS can provide vertical information on cloud structures such as double cloud layers or voids, ice distribution at different altitude levels, and information about the existence of particles of different sizes on different altitude levels that can complement the high horizontal resolution of the clouds from CIPS. Additionally, the combined dataset can be used to investigate how waves (inferred from albedo variations in CIPS) affect the cloud lifetime and how nucleation/sublimation processes affect the vertical distribution of cloud properties (inferred from a vertical cross section from OSIRIS).

To Abstract, p.1, line 16, the following text section has been added:

Important scientific questions on how PMC lifecycle is affected by changes in humidity and temperature due to atmospheric gravity waves, planetary waves and tides can be addressed by combining PMC observations in multiple dimensions. 2- and 3-dimensional cloud structures simultaneously observed by CIPS and tomographic OSIRIS

**provide a useful tool for studies of cloud growth and sublimation. Moreover, the combined CIPS/tomographic OSIRIS dataset can be used for studies of even more fundamental character, such as the question of the assumption of the PMC particle size distribution.**

**To p. 23, in beginning of section Discussion and Conclusions, the following sentence was added:**

**The analysis is performed for northern hemisphere 2010 and 2011 for a total set or 180 coinciding orbits at latitudes from 78N to 80N for local times ~ 15.45.**

**To p.23 line 6, the following text section has been added:**

**In this study, we have compared the PMC cloud properties cloud albedo and Ice water content from Odin OSIRIS limb tomography to the nadir viewing AIM CIPS. The analysis is performed for northern hemisphere 2010 and 2011 for a total set or 180 coinciding orbits at latitudes from 78N to 80N for local times ~ 15.45. The OSIRIS tomographic PMC dataset provides combined coarse horizontal and high vertical information, while CIPS provides preeminent horizontal PMC information. When combined in a common volume study, OSIRIS can provide vertical information of structures such as double cloud layers and ice voids as well as detailed particle size and number concentration information at various height levels to the detailed horizontal PMC information from CIPS. This information can be used to study how atmospheric waves of different scales (inferred from albedo variations in CIPS) alters the vertical distribution of cloud properties (inferred from OSIRIS). Additionally to such studies, the combined CIPS/tomographic dataset OSIRIS provide useful insight to more detailed studies of the PMC particle size distribution.**

- Page 6, line 20: This these -> These

       **Reply: Corrected**

- Page 7, line 28: Please consider adding a table that summarizes the uncertainties that have been mentioned in the text above. Ideally the corresponding uncertainties from CIPS should also be included to make this discussion easier to follow.

       **Reply: Good suggestion. Since we in this paper present the first thorough error characterization for OSIRIS tomography, a table specifying the uncertainties is in place. We added a table 1 to section 2.1.1 specifying OSIRIS uncertainties for PMC volume scatter coefficient and Ice Mass Density. However, since CIPS uncertainties is presented in a very thorough way in Lumpe (2013), we would like to not put CIPS uncertainties in the table, but instead point to this paper for more details.**

**Table 1. Summary of OSIRIS PMC uncertainties**

| Parameter | Description | Accuracy |
|---|---|---|
| PMC volume scatter coefficient β | Uncertainty estimated by Monte Carlo approach based on limb radiance uncertainty | ~10% |
| PMC Ice Mass Density | Uncertainty estimated by propagating uncertainty of β in 7 different wavelength regions through spectral analysis | ~10% |

- Page 9, line 28: Do you really mean systematic error or just systematic deviation/difference?

> **Reply: We mean systematic difference, corrected in text.**

– Figure 2: I am missing a discussion of the shape of this distribution. Is it coincidence that the distribution is roughly symmetric around 50%?

> **Reply: We regard the symmetry to be a coincidence. However, we added the following sentence about the shape to p.10 line 12.**
> **Fig. 2 shows an apparent dominance of CV that are either almost cloud-free (0%) or cloud-filled (100%). This is related to the choice of the size of the CV. If a larger CV would have been used, the distribution would not show such high numbers of detections for 0 and 100% clouds fraction.**

- Figure 3: Excellent!

> **Reply: We appreciate this comment!**

Page 13, line 7: Reference to Baumgarten et al., ACP 2010 should be added.

> **Reply: This reference has been added.**

- Page 13, line 17 : Please spell out "IMD".

> **Reply: This review comment made us aware of inconsistent use of Ice mass density and the abbreviation IMD, and also of Ice Water Content and the abbreviation IWC throughout the whole manuscript. We decided to introduce the abbreviation IWC in the abstract and Introduction (p.20 line 23) and only use IWC in the rest of the manuscript. The same has been adopted for Ice mass density for consistency. IMD is being first introduced in the abstract and then at p. 3 line 33, and used instead of ice mass density in the rest of the manuscript.**

- Figures 4, 6, and 7: haven't you regressed these data sets? I recommend to show the regression lines and indicate the corresponding parameters with their error bars.

**Reply: The datasets have been regressed. In the updated manuscript, we show the regression line (in blue) in figure 4, 6 and 7 (below) together with the parameters and error bars.**

[Figure]

Figure 4: Scatterplot of OSIRIS and CIPS observations of albedo in the common volume. The grey dashed line denotes the one-to-one line. The blue line is the regression line. The average Albedo for CIPS and OSIRIS for all the CV in the figure is indicated in the bottom right of the figure. OSIRIS error bars are a combination of systematic and statistical uncertainty. CIPS error bars are a combination of statistical uncertainty and uncertainty due to handling of dim cloud pixels (black), while the extended error bars (red) denote the uncertainty that is introduced by conversion factors that accounts in the difference in wavelength and scattering angle between the instruments. The reader is referred to the error discussion in the text for a more detailed description of CIPS and OSIRIS errors.

[Figure]

Figure 6: Scatterplot of OSIRIS and CIPS mean albedo in the CV. Same as Fig 4, but using thresholds on CIPS fill factor of 95% and OSIRIS scatter coefficient threshold of $10^{-10}$ m$^{-1}$sr$^{-1}$ as discussed in the text. The average albedo for CIPS and

OSIRIS for all the CV in the figure is indicated in the bottom right of the figure. The grey dashed line denotes the one-to-one line. The blue line is the regression line.

[Figure]

Figure 7: Scatterplot of OSIRIS and CIPS common volume ice water content. The average IWC for CIPS and OSIRIS is indicated in the bottom right of the figure. The error bars are a combination of the systematic and statistical uncertainty from each instrument, thus representing a total uncertainty that is relevant when comparing the datasets. The grey dashed line denotes the one-to-one line. The blue line is the regression line.

- Figure 4: Why is the OSIRIS error bar increasing with OSIRIS albedo?

> **Reply: The OSIRIS albedo error in Fig. 4 is a combination of the systematic error caused by calibration and the estimated error due to the vertical integration of scattering coefficient to albedo. For a faint and thin cloud, there is often only a few vertical levels contribute to the error in the vertical integration thus the combined error is small. However for a bright cloud with larger vertical extent a *larger number of levels* contribute to the error, and therefore the combined error becomes larger.**

- Page 19, line 15/16: clearly three examples from such a large data set cannot be representative. At most they may illustrate the class of comparisons. Please reword.

> **Reply: This is true. We changed the sentence on line 15/16: "These orbits were chosen to illustrate both orbits when the instruments show good agreement and when the instruments disagree and are thus representative for the total set of orbits available for this study",**

**to**

**"These particular orbits were chosen to illustrate some examples of when the clouds in the CV show good agreement, and point out some example when the cloud observations in the CV disagree, and thus illustrate for the reader the range of cloud observations available for this study."**

- Page 19, line 22: that are we have -> that we have

**Reply: Corrected.**

**Additional changes to the paper:**

**We noted that CIPS had not been referenced when it was first mentioned in the Introduction, and therefore we added the reference (Russel et al, 2009) on p.2 line 16.**

**We also noted that OSIRIS had not been referenced properly when first mentioned, and therefore changed the sentence on page 3, line 10 : "The OSIRIS PMC retrieval for the normal limb scans assumes that the PMC layer is spatially homogeneous along the instrument line of sight (LOS).**

**to:**

**"The PMC retrieval for the limb-viewing Optical Spectrograph and Infrared Imager System (OSIRIS) (Llewellyn et al., 2004) on the Odin satellite (Murtagh et al., 2002) assumes that the PMC layer is spatially homogeneous along the instrument line of sight (LOS) for the normal limb scans."**

**Reply to comments by reviewer #2**

**The comments from the reviewer are written in standard font. The comments from the authors are written in bold font. All page numbers and line numbers refer to the original draft.**

General Comments

This paper presents a detailed comparison of polar mesospheric cloud (PMC) albedo and ice water content between two different satellite instruments. The instruments are the Optical Spectrograph and InfraRed Imager System (OSIRIS) on the Odin satellite and the Cloud Imaging and Particle Size (CIPS) instrument on the Aeronomy of Ice in the Mesosphere (AIM) satellite. Because OSIRIS typically views PMCs on the limb whereas CIPS typically views PMCs in the nadir, the authors have carefully considered coincidence criteria, scattering conditions, observation geometry, and instrument sensitivity in the uniquely coordinated study between the two instruments. As part of the study, the authors present the first thorough error characterization of OSIRIS tomographic cloud brightness and ice water content.

This is an important paper and establishes a valuable precedent for subsequent comparisons between PMC limb viewing instruments and PMC nadir imagers. The results show good agreement, particularly given the diversity of data included in the study. Most importantly, however, the authors provide an exhaustive error analysis that will be a useful reference in future PMC correlative studies.

The Reviewer recommends the paper for publication provided that the authors address the comments below. The "Specific Comments" are relatively minor but important, particularly in providing context of their results with the existing body of work on this topic.

> **Reply: We thank the reviewer for this very kind comment. We also want thank the reviewer for providing much valuable advice and recommendations of how to improve the manuscript. Your specific comments and ideas for the section "Conclusions" is something we are very grateful for.**

Specific Comments

(1). Abstract. Please indicate latitude range and years used in the analysis. Also, if PMC frequency is not compared between OSIRIS and CIPS within the common volume, the authors should explicitly say so in the abstract.

> **Reply: We have added this information to the abstract.**

(2). p. 8. Lines 9-11. Did Benze et al. [2011] use the operational CIPS product to compare directly with SBUV? The Reviewer looked at this paper and it appears that the good agreement with SBUV as stated here arises because a separate CIPS retrieval was developed to simulate the SBUV PMC retrieval. This is not a validation of the CIPS or SBUV data, which is what is suggested by this statement. How do operational SBUV and v4.20 CIPS PMC albedos, IWC and frequencies compare for the same volume and the same time at these high latitudes (78-80 N)? If the authors do not have a ready answer or if it is beyond the scope of this work then they should be explicit about what was done previously to find agreement between CIPS and SBUV (i.e. a separate CIPS algorithm). They could also delete these sentences entirely without loss of content to the paper.

**Reply: Good point. As the reviewer writes, Benze et al (2011) used a separate "SBUV-type" algorithm. To make this clear in the text, we changed the lines 9-11 on p. 8 to:**

**CIPS cloud detections and albedo values were previously compared to the solar backscatter ultraviolet (SBUV/2) instruments [Benze et al., 2009; 2011]. This was accomplished by applying a "SBUV-type" algorithm to the CIPS level 1A data to make the two datasets comparable. Cloud frequency and brightness from CIPS were shown to be in good agreement with SBUV/2 retrievals.**

**We do not know how SBUV and CIPS v4.20 compare for the same volume and time, and it is beyond the scope of this work to analyze this.**

(3). p. 16, end of section. Please include a paragraph here explicitly indicating what is done with pixels where there are no clouds at all. If the authors have set all pixels less than 2e-6 sr-1 to zero, then they need to explicitly say here whether they have averaged the zeros into their calculations of albedo and IWC or not. This distinction has historically been a source of great confusion in the field of PMCs. It may be that at these high latitudes there is always a cloud within the common volume for their limited dataset and if so they need to say that as well. This does not appear to be the case from looking at Figure 2. However, the authors winnow the dataset to 788 total observations (p. 17, line 9) so it is not clear how Figure 2 evolves with the study.

**Reply: The zero pixels are included in the average. The reviewer is right that there is often cloud within the CV element in our combined dataset, but not always. As can be seen by figure 2, for several hundred observations CIPS observe no cloud in the CV element, and for many observations CIPS observe only partly cloudy pixels in the CV element. As suggested by the reviewer, we add a section in the end of p. 16 to discuss this. The following text section has**

**been added:**

**To summarize, the following two sensitivity adjustments are applied to CIPS**

**pixels: (1) All pixels with an albedo < 2e-6 sr$^{-1}$ are set to zero, and the zeros are included in the horizontal average of CIPS albedo (2) A filling factor of 95% is required, meaning that only observations where CIPS observe at least 95% cloudy pixels in the CV element are included in the comparison.**

**To make this clear also earlier in the paper, we added the following sentence to section 3.3, p. 13, line 14:**

**As noted in section 2.2.1, we manually set all dim CIPS pixels < 2e-6 sr$^{-1}$ to zero in the qualitative comparison. These zero pixels have been included in the average of CIPS mean albedo.**

(4). Conclusions. This section is lacking a summary of relevant conditions under which the comparisons are made. This includes (but is not limited to) the years studied, the latitudes used and the local times of the comparisons. This should also emphasize that the authors are comparing albedo and IWC and not PMC frequency. This section is also lacking a summary of previous related work by Bailey et al. [2015] using common volume observations of SOFIE and CIPS on the same AIM satellite. Bailey et al. state that CIPS IWC is a factor of two smaller than SOFIE IWC, differences that are generally larger and go in the opposite direction of the present work with OSIRIS. Although the authors have done a thorough analysis of their two datasets (OSIRIS and CIPS), the reader should be made aware of these differences of CIPS IWC with the limb viewing SOFIE IWC. This is all the more important because IWC is the native measurement quantity for SOFIE. Differences in the method of observation, calibration, coincidence criteria, latitudes of the comparison, the years studied, the solar zenith angles and the local times of the comparison may all play a role in reconciling these differences and could be included to raise awareness with the reader. The above could be done with two paragraphs and if the authors prefer they could rename this section "Discussion and Conclusions".

**Reply: The authors agree, the difference in IWC compared to Bailey et al., (2015) needs to be addressed. We are aware of the differences between our CIPS/OSIRIS study and the CIPS/SOFIE study of Bailey et al. In our study we compare two instrument that have similar measurement technique. They both measure scattered light in the ultraviolet from a common volume, and therefore a direct comparison between CIPS and Osiris IWC is instructive. Despite this, the observed difference between the CIPS and SOFIE remain.**

**We changed the title of the section to "Discussion and Conclusions" and added the following paragraph to p. 23, line 14:**

**A reason for why OSIRIS IWC is biased low to CIPS IWC in is considered to arise from how ice mass density (IMD) is calculated in OSIRIS PMC retrieval. When calculating IWC from the vertical integration of OSIRIS IMD data, we currently only take into account retrieval pixels that are bright enough so that a spectroscopic size retrieval is feasible. It is possible that the OSIRIS analysis**

misses ice from weak pixels where mean radii smaller than 20 nm are reported. Such weak cloud pixels cover typically an altitude range of 1 km in the upper part of the cloud. A rough estimate of how much ice OSIRIS misses by ignoring such pixels can be given by considering how much ice can be produced when converting a typical concentration of water vapour at 86 km (e.g., 3 ppm water vapour in 2e14 cm$^{-3}$ air) into ice. Calculating this, a 1 km thick layer contributes to the overall IWC with 18 g km$^{-2}$. This is more or less the difference that we see in Figure 7 between OSIRIS and CIPS IWC. Bailey et al. (2015) compare albedo, radius, and IWC in a CIPS-SOFIE CV that is located at high SZAs. These clouds occur right at the edge of visibility, where the Rayleigh background is small and has a large slope vs. SZA. This is a difficult region to observe PMCs, as even a small error in the CIPS Rayleigh background is expected to have a large impact on the retrieved cloud products. Bailey et al. (2015) apply a justifiable correction to the CIPS background removal which brings the SOFIE and CIPS observations, specifically albedo and IWC, into much better agreement than without (their Figure 11). The present study uses cloud observations at a much more favorable SZA range (59-71 deg). Here, CIPS background changes such as found by Bailey et al. (2015) are negligible, and no CIPS background correction is necessary. Bailey et al. (2015) show that CIPS IWC is ~30% smaller than SOFIE IWC (their Figure 11). In the present study, CIPS suggests significantly more IWC than OSIRIS in the common volume. This difference between the two studies is important because IMD is the native measurement quantity for SOFIE, which can thus be seen as a reference measurement. SOFIE uses the technique of satellite solar occultation to measures vertical profiles of limb path atmospheric transmission, and as such is much more sensitive to small ice particles than OSIRIS. The two studies use different methods: Bailey et al. (2015) model what CIPS should observe based on SOFIE observations, whereas this study corrects for sensitivity differences and compares the horizontally and vertically averaged quantities. Another difference is that this study uses observations at more favorable SZAs and higher latitudes, therefore avoiding uncertainties of the CIPS background and containing generally brighter clouds with more ice.

To p. 24, in beginning of section Discussion and Conclusions, the following sentence was added:

The analysis is performed for northern hemisphere 2010 and 2011 for a total set or 180 coinciding orbits at latitudes from 78N to 80N for local times ~ 15.45.

To p. 24, line 21, the following sentence was added:

Due to the limitation of a total of 18 days of observations during the seasons NH 2010 and NH 2011, we have not performed a detailed comparison of cloud frequency.

Technical Corrections

p. 1, line 17. "ice" should be "ice water content". Similarly, on line 20 "ice content" should be "ice water content" to avoid confusion.

**Reply: This has been corrected.**

p. 2, line 17. "larg" should be "large"

**Reply: Corrected.**

p. 2, line 30. ". . .reason for the increasing visibility of PMCs at mid-latitudes" should be ". . .reason for the increasing visibility of PMCs at mid-latitudes in the modern era." To the Reviewer's knowledge, decadal-scale trends of mesospheric clouds observed from the ground since the late 20th century are weak or non-existent.

**Reply: This is certainly true, corrected.**

p. 2, line 36. "advantage" should be "advantages".

**Reply: Corrected.**
* * *
p. 10, Figure 1. This figure has a geographic range that is much larger than the region of interest and could be improved dramatically. On lines 9-10 the authors say that the cloud albedo is variable but the region of interest is drawn over the data so the reader cannot see this. By reducing the latitude and longitude ranges of the image, only the borders of the region of interest can be drawn and the boxed region can remain unfilled so that the reader can see the structure within. If the geographic range is small enough, the red area could also be drawn with borders rather than filled. If the authors prefer, the figure could be drawn with two panels: Panel "a" could be the current figure and panel "b" could be the zoomed in version. Please also include a color bar showing the range of cloud albedo in the figure(s).

**Reply: Really good point. We did as the reviewer suggested and updated the figure to two panel plot, where the lower panel is a zoomed in version. Instead of a red box, we mark each pixel in the CV element with thin grey lines. The figure text has been updated and a color bar added.**

[Figure]

**Figure 1: Example orbit showing a CIPS/OSIRIS coincidence on a polar map plot for CIPS orbit 50777 and OSIRIS orbit 17098. The top panel shows the CIPS orbit strip. The white line on top of the CIPS orbit strip indicates the overlapping ~660 CIPS pixels in the CV. Each CV is composed of ~10 CV elements. The bottom panel shows an example of only those 66 pixels contained within one CV element.**

p. 11, Figure 2. Please include tick marks on the x and y axes to better guide the reader. Also, please indicate the total number of detections either within the Figure or in the caption. Since this is the first figure quantitative showing CIPS data, it would also be instructive to indicate that this is CIPS data, and include average latitude, local time, year of the data and a CV frequency either within the Figure or in the caption. Thank you.

**Reply: The figure and caption has been updated according to request. We also corrected the number of detections on p. 10 line 10 from 1420 to 1513.**

[Figure]

**Figure 2: AIM CIPS occurrence of PMCs in the common volume with OSIRIS during the observations in 2010-2011. The plot shows the number of CVs containing a certain fraction of CIPS pixels with identified PMCs. Average latitude is 80° N, local time ~15.45. The mean cloud fraction is 62%.**

p. 13, line 15. Please include here the ranges of Cspectral and Cphase used in the analysis so that the reader can appreciate the impact of these adjustments in the context of the data.

**Reply: Good point. On p. 13, line 9 the following section describing the ranges of the of the conversion factors has been added:**

**The spectral conversion factor $C_{phase}$ depend on the solar scattering angle and increases with increasing particle size. For particles in the range 1-20 nm $C_{phase}$ varies between 1.0-1.5, for 21-50 nm $C_{phase}$ varies between 0.6-2.7 and for particles in the range 51-100 nm $C_{phase}$ varies between 0.4-5.6. Note that the conversion factors given here range over a large range of solar scattering angles, for a single given solar scattering angle the range of $C_{phase}$ is much more limited. The spectral conversion factor $C_{spectral}$ range between 0.8 to 1.0.**

p. 14. The offset and uncertainty in line 17 is a bit different than line 26. This is further modified on p. 17 line 15, but is still not quite the same as reported in the abstract and summary. Please check to make sure the numbers self-consistent throughout. Thank you.

**Reply: We thank the reviewer for noticing this. In p.14, line 17 and line 26, and p.17 line 15, the offset and uncertainty should off course be the same and is 2.8e-6 sr$^{-1}$ (±2.4e-6 sr$^{-1}$). This has been corrected in the manuscript. This result**

**refer to the albedo comparison in the first step of the analysis before we adjust for differences in sensitivity and without restriction on CIPS fill factor.**

**The offset uncertainty reported in the abstract and summary (3.4e-6 sr$^{-1}$ (± 2.9e-6 sr$^{-1}$)) is the result from the albedo comparison after an adjustment for sensitivity (described on p. line) and threshold on CIPS fill factor (described on p. line) has been applied, and therefore differ from p. 14 and p.17.**

p. 15, Figure 4 caption. "error bars is" should be "error bars are" and "error bar denote" should be "error bars denote".

**Reply: Corrected.**

p. 16 lines 2-4. Do the authors mean the error bars in Figures 4 and 6? Please indicate the figures explicitly. Also, the Reviewer only sees black (not grey) error bars in these figures. Are they referring to these? Please be explicit. Thank you.

**Reply: Yes, we mean 4 and 6, and the sentence has been corrected. The grey color bars are too dark to be called grey, and we update the text to "black" instead.**

p. 20, line 30. A wind of 100 m/s near 85 km seems large. Can the authors provide a reference for this or otherwise justify this wind speed? Is it possible that the cloud could be sublimating and reforming elsewhere? If so the authors should state that as a possibility. To this end, the authors should include the time difference between the two observations in the captions of Figures 8, 9 and 10.

**Reply: For reference of the 100 m/s wind speed, the authors referred to the master study by Kåre Backer-Owe from Norwegian University of Science and Technology," Behavior of the S1 and S2 Components of the Semidiurnal Tide in the MLT" where observations of wind speeds of 100 m/s are observed by the Super Dual Auroral Radar Network (SuperDARN) for the station Dragwoll at 63N at 82 km altitude. (fig 2.2)**

https://ntnuopen.ntnu.no/ntnu-xmlui/handle/11250/2398890

**For wind velocities at higher latitudes, the study by Stober et al (2013) is probably a better source to site. In this paper, wind velocities of ~50 m/s are observed by the Middle Atmosphere Alomar Radar System (MAARSY).**
**We cannot exclude the possibility that part of the cloud could be sublimating during time of observation, and agree that this is a good point to mention.**

**We update the text section staring on p.20, line 30 to:**

**At PMC altitude the horizontal wind is mainly modulated by atmospheric tidal**

**waves and gravity waves. Horizontal wind velocities of ~50 m/s have been observed by The Middle Atmosphere Alomar Radar System (MAARSY) in the northern Norway (Stober et al., 2013). In this CV, the time difference between CIPS-OSIRIS observations is ~5 minutes During this time, a wind speed of 50 m/s would transport a cloud 15 km, which corresponds to the half width of OSIRIS LOS. It is likely that the cloud region outside the CIPS-OSIRIS CV (as indicated by the pink lines in the third panel) could have been transported into the CV by wind directed across the OSIRIS LOS**.

p. 24, lines 25-27. Please explicitly indicate the version of CIPS data used in this study here (in addition to p. 8, line 8).

**Reply: This has been added.**

Figures 4, 6 and 7. Please indicate explicitly whether null detections are included in the indicated average. If null detections are ignored in these comparisons then they should say that instead.

**Reply: In figure 4, 6 and 7, null detections are included in the average. This clarification has been noted by adding a text section describing this in p.16. To make it clear to the reader, the caption of fig 4,6 and 7 has also been updated with this information.**

[revised manuscript text omitted]

**Borttaget:** ice mass density

**Borttaget:** ice mass density

**Borttaget:** ice water content

**Borttaget:** ice mass density

**Borttaget:** ice water content

**Borttaget:** . Taking into account that tThe mean CIPS IWC error strongly increases with decreasing particle size, . As described in Section 2.2.1, we take this into account by screening out the suspicious IWC detections, by only including CIPS pixels that report a particle radius > 20 nm when we calculate the mean IWC in each CV element. For consistency with the albedo comparison, we have added applied a CV fill threshold of 95%. For OSIRIS, the retrieval threshold (10$^{-10}$ m$^{-1}$sr$^{-1}$) is applied in the vertical integration to form ice water content in the common volume by vertically integrating ice mass density at each level. We find an offset between OSIRIS and CIPS IWC, and we quantify this offset (OSIRIS - CIPS) to -22 g km$^{-2}$ ±14 g km$^{-2}$. The error bars in Figure 7 represent a total error that is relevant for the comparison between the instruments by combining systematic and statistical error from each dataset. Based on the discussion in section 2.1.1, the OSIRIS errorbars represent the combination of combine the absolute uncertainty of ice mass density due to the 10 % measurement accuracy and a random statistical uncertainty that is obtained by propagating the uncertainty of the scattering coefficient through the derivation of the ice mass density at each level. The resulting error in ice water content in the common volume element is calculated in the column integration as the combined error of all vertical levels. Similarly, The the CIPS errorbars (in grey) represent the mean and statistical uncertainty propagated from all individual CIPS pixels in the CV element. The individual uncertainty for each pixel is taken directly from Fig. 21 in Lumpe et al. (2013) where the estimated uncertainty of CIPS ice water content is 
[revised manuscript text omitted]

| **Sida 32: [2] Borttaget** | **Lina Broman** | **2019-01-24 17:15:00** |
|---|---|---|

Thomas, G.: Is the polar mesosphere the miner's canary of global change?, Adv. Space Res., 18(3), 149–158, 1996.

Thomas, G. E.: Are noctilucent clouds harbingers of global change in the middle atmosphere? Adv. Space Res., 32, 1737–1746, 2003.

---

## Author Response (AR2)

**Response to Reviewer**

**We thank the reviewer for the helpful comments and suggestions. Our responses are following reviewer comments in bold.**

Review of "Common volume satellite studies of polar mesospheric clouds with Odin/OSIRIS tomography and AIM/CIPS nadir imaging" by Broman et al.
The Reviewer thanks the authors for carefully answering the Reviewer's concerns. There is one remaining important point and that is in the discussion of the Bailey et al. [2015] study on pp. 25-26 comparing CV SOFIE and CIPS IWC. The specific comments on that paragraph are as follows:

1. On p. 25 line 33 the authors use the term "edge of visibility" in regards to the CIPS PMC observations in the CV with SOFIE. This is a little misleading because the authors have just described how OSIRIS cannot see small particles, which seems to be a different problem than what the authors are getting at here. The reviewer suggests "at the terminator" instead of "at the edge of visibility" if that is what they mean, to avoid any confusion.

> **Reply: We agree, this is misleading. The manuscript has been updated according to request.**

2. On p. 25 line 36 the improvement is better but it seems hard to say "much" better when CIPS IWC goes from 50% less to 30% less compared to SOFIE. The authors do not lose anything here if they delete "much".

> **Reply: This is true. We deleted "much" as suggested.**

3. The authors should clarify that the adjustments that Bailey et al. [2015] made to the CIPS background are not in the operational CIPS v4.2 product. Without any adjustments at all the Bailey et al. abstract states that the CIPS IWC is 50% less than SOFIE in the CV. The Reviewer therefore suggests that the sentence beginning on p. 25 line 39 should read something like: "Bailey et al. (2015) show that the reported v4.2 CIPS IWC is 50% smaller than the SOFIE IWC in the common volume, but only ~30% smaller when correcting the CIPS background removal (their Figure 11)." The authors should be aware that later versions of the CIPS product (i.e. v5) are supposed to correct this bias, though to the Reviewer's knowledge this has not yet been verified.

> **Reply: Yes, this should be pointed out to avoid confusion. We updated the manuscript with your suggested new sentence.**

4. On line 39-40 the Reviewer suggests the authors be more quantitative in this summary as they have just quoted the 30% difference in the sentence before. From Figure 7 it appears that the average CIPS IWC is 33% more than that of OSIRIS in the CV. If that is the correct number, then the Reviewer suggests rewriting the sentence beginning at the end of line 39 as follows: "In the present study, we find that the CIPS IWC is 33% greater than the OSIRIS IWC in the common volume (Figure 7)." If they would like to add uncertainties to this number to be consistent with the abstract then they should.

> **Reply:  Good suggestion. On average, the CIPS IWC is 33% greater, that is correct. The OSIRIS IWC uncertainties of 14 g km$^{-2}$ correspond to 22%. We rewrote the sentence to:**
>
> **"In the present study, we find that the CIPS IWC is 33% (±22%) greater than the OSIRIS IWC in the common volume (Figure 7)."**

[revised manuscript text omitted]